# Homeostatic synaptic plasticity of miniature excitatory postsynaptic currents in mouse cortical cultures requires neuronal *Rab3a*

Andrew G Koesters[1]*, Mark M Rich[2], Kathrin Engisch[3]

[1]Department of Pharmacology, Physiology, and Neurobiology, University of Cincinnati College of Medicine, Cincinnati, United States; [2]Department of Neuroscience, Cell Biology and Physiology, Boonshoft School of Medicine, Wright State University, Dayton, United States; [3]Department of Neuroscience, Cell Biology and Physiology, Boonshoft School of Medicine and the College of Science and Mathematics, Wright State University, Dayton, United States

*For correspondence: koesteag@ucmail.uc.edu

Competing interest: The authors declare that no competing interests exist.

## eLife Assessment

This **valuable** study presents findings on the role of the small GTPase Rab3A in homeostatic plasticity. While the study provides **solid** evidence for a requirement of Rab3A in homeostatic up-scaling in cultured mouse neurons, it does not provide a model of how Rab3A is involved in homeostatic plasticity. The work will be of interest to researchers in the field of synaptic transmission and synaptic plasticity.

**Abstract** Following prolonged activity blockade, amplitudes of miniature excitatory postsynaptic currents (mEPSCs) increase, a form of plasticity termed 'homeostatic synaptic plasticity'. We previously showed that a presynaptic protein, the small GTPase *Rab3a*, is required for full expression of the increase in miniature endplate current amplitudes following prolonged blockade of action potential activity at the mouse neuromuscular junction (NMJ) in vivo, where an increase in postsynaptic receptors does not contribute. It is unknown whether this form of *Rab3a*-dependent homeostatic plasticity at the NMJ shares any characteristics with central synapses. We show here that homeostatic synaptic plasticity of mEPSCs is impaired in mouse cortical neuron cultures prepared from *Rab3a*⁻/⁻ and mutant mice expressing a single-point mutation of *Rab3a*, *Rab3a Earlybird* mice. To determine if *Rab3a* is involved in the well-established homeostatic increase in postsynaptic AMPA-type receptors (AMPARs), we performed a series of experiments in which electrophysiological recordings of mEPSCs and confocal imaging of synaptic AMPAR immunofluorescence were assessed within the same cultures. We found that the increase in postsynaptic AMPAR levels in wild-type cultures was more variable than that of mEPSC amplitudes, which might be explained by a presynaptic contribution, but we cannot rule out variability in the measurement. Finally, we demonstrate that *Rab3a* is acting in neurons because only selective loss of *Rab3a* in neurons, not glia, disrupted the homeostatic increase in mEPSC amplitudes. This is the first demonstration that a protein thought to function presynaptically is required for homeostatic synaptic plasticity of quantal size in central neurons.

## Introduction

One of the most studied phenomena triggered by prolonged activity blockade is the increase in amplitudes of miniature excitatory postsynaptic currents (mEPSCs) in neurons. First demonstrated in cultures of dissociated cortical neurons (*Turrigiano et al., 1998*) and spinal cord neurons (*O'Brien et al., 1998*), the compensatory response to prolonged loss of activity has been dubbed 'homeostatic synaptic plasticity' (*Turrigiano and Nelson, 2004*; *Pozo and Goda, 2010*). These first two studies provided evidence that glutamate receptors were also increased after prolonged inactivity, via an increased response to exogenously applied glutamate (*Turrigiano et al., 1998*) and increased immunofluorescent labeling of GluA1 receptors at synaptic sites (*O'Brien et al., 1998*). The increase in synaptic AMPA receptors after activity blockade has been confirmed with immunofluorescence in multiple studies of hippocampal and cortical cultures treated with TTX, including (*Wierenga et al., 2005*; *Hou et al., 2008*; *Ibata et al., 2008*; *Jakawich et al., 2010*; *Tatavarty et al., 2013*; *Xu and Pozzo-Miller, 2017*; *Dubes et al., 2022*). It is now well accepted that the homeostatic increase in mEPSC amplitudes in neurons is due to an increase in postsynaptic AMPA receptors.

In our previous work, we found that a TTX cuff applied for 48 hr around the sciatic nerve in mice led to an increase in the amplitude of the miniature endplate currents (mEPCs) recorded in tibialis anterior muscles. Surprisingly, and unlike at central synapses, we could find no evidence that the acetylcholine receptor levels at the neuromuscular junction (NMJ) were increased in the TTX-cuff treated mice (*Wang et al., 2005*). This result led us to search for presynaptic molecules that might homeostatically regulate the presynaptic quantum. In previous studies in chromaffin cells, we identified the small GTPase *Rab3a*, a presynaptic vesicle protein, as a regulator of synaptic vesicle fusion pore opening (*Wang et al., 2008*), so we examined whether deletion of *Rab3a* (*Rab3a⁻/⁻*) might prevent homeostatic upregulation of mEPC amplitude at the NMJ. The results were clear: the homeostatic increase in mEPC amplitude induced by the TTX cuff was strongly reduced in the *Rab3a⁻/⁻* mouse and was completely abolished in the *Earlybird* mutant (*Rab3aEbd/Ebd*), which has a single-point mutation in *Rab3a* that causes a shift toward early awakening due to a shorter circadian period, that is more dramatic than that of the *Rab3a* deletion mouse (*Kapfhamer et al., 2002*; *Wang et al., 2011*). These results led us to conclude that the homeostatic increase in mEPC amplitude at the NMJ is a presynaptic phenomenon.

It has remained somewhat of a mystery what the role of *Rab3a* is in synaptic transmission. The *Rab3a⁻/⁻* mouse has minimal phenotypic abnormalities, with evoked synaptic transmission and mEPSCs essentially normal in hippocampal slices (*Geppert et al., 1994*). At the *Rab3a⁻/⁻* NMJ, reductions in evoked transmission were detected, but only under conditions of reduced extracellular calcium (*Coleman et al., 2007*). There are some modest changes in short-term plasticity during repetitive stimulus trains, with increased depression observed in response to moderate frequencies in hippocampal slices (*Geppert et al., 1994*) and at the mammalian NMJ (*Coleman and Bykhovskaia, 2009*), but increased facilitation in response to high frequency stimulation at the mammalian NMJ (*Coleman and Bykhovskaia, 2010*). The most dramatic effect of loss of *Rab3a* is the abolishment of a presynaptic form of long-term potentiation at the mossy fiber-CA3 synapse that is induced by 25 Hz stimulation in the presence of NMDA blockers (*Weisskopf et al., 1994*; *Castillo et al., 1997*). Our result that prolonged inactivity-induced plasticity of mEPC amplitude at the mammalian NMJ is disrupted with loss of function of *Rab3a* further cements a role for *Rab3a* in activity-dependent plasticity of synaptic strength. However, a crucial question that remains is whether the TTX-cuff induced effect on the amplitude of the spontaneous synaptic current at the NMJ shares underlying mechanisms with central synapses. We set out to answer this question for the more well-studied phenomenon of homeostatic plasticity of mEPSCs in dissociated mouse cortical neurons prepared from wild-type, *Rab3a⁻/⁻* mice and *Rab3aEbd/Ebd* mice treated with TTX for 48 hr. We show here that homeostatic synaptic plasticity of mEPSC amplitude was strongly reduced in cultures from *Rab3a⁻/⁻* mice and completely abolished in cultures from *Rab3aEbd/Ebd* mice, supporting a shared mechanism with the NMJ.

## Results

We previously reported that mixed cultures of cortical neurons and glia prepared from postnatal day 0 to 2 mouse pups responded to a block of action potential-mediated activity by a 48-hr TTX treatment with an increase in mEPSC amplitude (*Hanes et al., 2020*). Here, we asked the question

whether cortical cultures prepared from mice lacking the small GTPase *Rab3a* or expressing a point mutation of *Rab3*, *Rab3a* Earlybird, have an altered homeostatic plasticity response to a prolonged loss of network activity. The *Rab3a* Earlybird mutation was discovered in a mutagenesis screen in mice for genes involved in the rest–activity cycle (**Kapfhamer et al., 2002**). In that study, the authors concluded that the Earlybird mutation was likely dominant-negative, as the shift in circadian period was substantially more dramatic than that observed in the *Rab3a*$^{-/-}$ mice. As described in the Introduction, the effects of loss of *Rab3a* have been modest, so we included experiments using the *Rab3a* Earlybird mutant mice to see if there would be a more robust phenotype, and in doing so, strengthen our conclusions based on the *Rab3a* deletion mice. To obtain *Rab3a*$^{-/-}$ and *Rab3a*$^{Ebd/Ebd}$ homozygotes, we established two mouse colonies of heterozygous breeders with cultures prepared from pups derived from a final breeding pair of homozygotes. Although we backcrossed *Rab3a*$^{+/-}$ with *Rab3a*$^{+/Ebd}$ for 11 generations, clear differences in mEPSC amplitudes in untreated cortical cultures for the two wild-type strains (see below) and in calcium current amplitudes in adrenal chromaffin cells for the two wild-type strains (unpublished obs.) remained. Therefore, throughout this study, there are two *Rab3a*$^{+/+}$ or 'wild-type' phenotypes.

We found that loss of *Rab3a* greatly impaired the homeostatic increase after activity blockade. Example current traces of spontaneously occurring mEPSCs recorded from pyramidal neurons in untreated (CON) 13–14 DIV cortical cultures and sister cultures treated with 500 nM TTX for 48 hr prepared from wild-type and *Rab3a*$^{-/-}$ mice in the *Rab3a*$^{+/-}$ colony are shown in **Figure 1A and B**, respectively. Average mEPSC waveforms from the same recordings are shown in **Figure 1C, D**. The mean mEPSC amplitudes for 30 control and 23 TTX-treated neurons from *Rab3a*$^{+/+}$ cultures are displayed in the box and whisker plot in **Figure 1E**; after activity blockade, the average mEPSC amplitude increased from 13.9 ± 0.7 to 18.2 ± 0.9 pA. In contrast, in cultures prepared from *Rab3a*$^{-/-}$ mice, the average mEPSC amplitude was not significantly increased, for 25 untreated cells and 26 TTX-treated cells (**Figure 1F**, 13.6 ± 0.1 vs. 14.3 ± 0.6 pA). We found that TTX treatment also resulted in an increase in mEPSC frequency in cultures prepared from *Rab3a*$^{+/+}$ mice, as shown in **Figure 1G** (CON, 2.26 ± 0.37 s$^{-1}$; TTX, 4.62 ± 0.74 s$^{-1}$). This TTX-induced frequency increase was strongly reduced in cultures from *Rab3a*$^{-/-}$ mice, but there were a few outliers with very high frequencies still present after TTX treatment (**Figure 1H**, CON, 2.74 ± 0.49 s$^{-1}$; TTX, 3.23 ± 0.93 s$^{-1}$).

We next examined homeostatic plasticity in cultures prepared from wild-type mice in the *Rab3a*$^{+/Ebd}$ colony. Example current traces of mEPSCs, and average mEPSC waveforms, are shown in **Figure 2A and C**, respectively, for cortical cultures prepared from wild-type mice in the *Rab3a*$^{+/Ebd}$ colony. Treatment with TTX for 48 hr led to a significant increase in the average mEPSC amplitude of 23 TTX-treated cells when compared to 20 untreated cells (**Figure 2E**, CON, 11.0 ± 0.6 pA; TTX, 15.1 ± 1.2 pA). We note here that while the two wild-type strains respond very similarly to activity blockade, the mean mEPSC amplitude in untreated cultures was different between the two strains, 13.9 ± 0.7 pA, wild type from *Rab3a*$^{+/-}$ colony; 11.0 ± 0.6 pA, wild type from *Rab3a*$^{+/Ebd}$ colony, but this difference did not reach statistical significance in a Tukey means comparison after a two-way ANOVA (p = 0.27).

We found a complete disruption of homeostatic plasticity in cortical cultures prepared from *Rab3a*$^{Ebd/Ebd}$ mice, as can be seen in viewing example mEPSC traces and average mEPSC waveforms (**Figure 2B and D**, respectively). The lack of TTX effect was confirmed in a comparison of mEPSC amplitude means for 21 untreated and 22 TTX-treated cells (**Figure 2F**, CON, 15.1 ± 1.0 pA vs. TTX, 14.6 ± 1.1 pA). There was a trend toward increased mEPSC frequency in TTX-treated cultures from this wild-type strain, but the difference was not as robust as for the wild-type cultures from the *Rab3a*$^{+/-}$ colony and did not reach statistical significance (**Figure 2G**, CON, 1.15 ± 0.19 s$^{-1}$; TTX, 2.54 ± 0.55 s$^{-1}$). This trend was not observed in neurons from cultures prepared from *Rab3a*$^{Ebd/Ebd}$ mice, likely due to an increase in frequency in untreated cells—frequencies remained high in TTX-treated cells (**Figure 2H**, CON, 1.71 ± 0.41 s$^{-1}$; TTX, 3.05 ± 0.80 s$^{-1}$). Our results show that the homeostatic increase in mEPSC amplitude after activity blockade is disrupted in both *Rab3a*$^{-/-}$ and the *Rab3a*$^{Ebd/Ebd}$ cortical neurons, strongly supporting a crucial role of functioning *Rab3a* in this process.

The disruption in homeostatic plasticity differs for *Rab3a*$^{-/-}$ and *Rab3a*$^{Ebd/Ebd}$. In the *Rab3a*$^{-/-}$ dataset, mEPSCs from untreated cultures were indistinguishable from mEPSCs from *Rab3a*$^{+/+}$ untreated cultures, demonstrating loss of *Rab3a* has no impact on basal activity mEPSC amplitudes. In the *Rab3a*$^{Ebd/Ebd}$ dataset, mEPSC amplitudes from untreated cultures were significantly larger than those of untreated cultures from wild-type mice, as can be seen comparing the CON values in **Figure 2E, F**

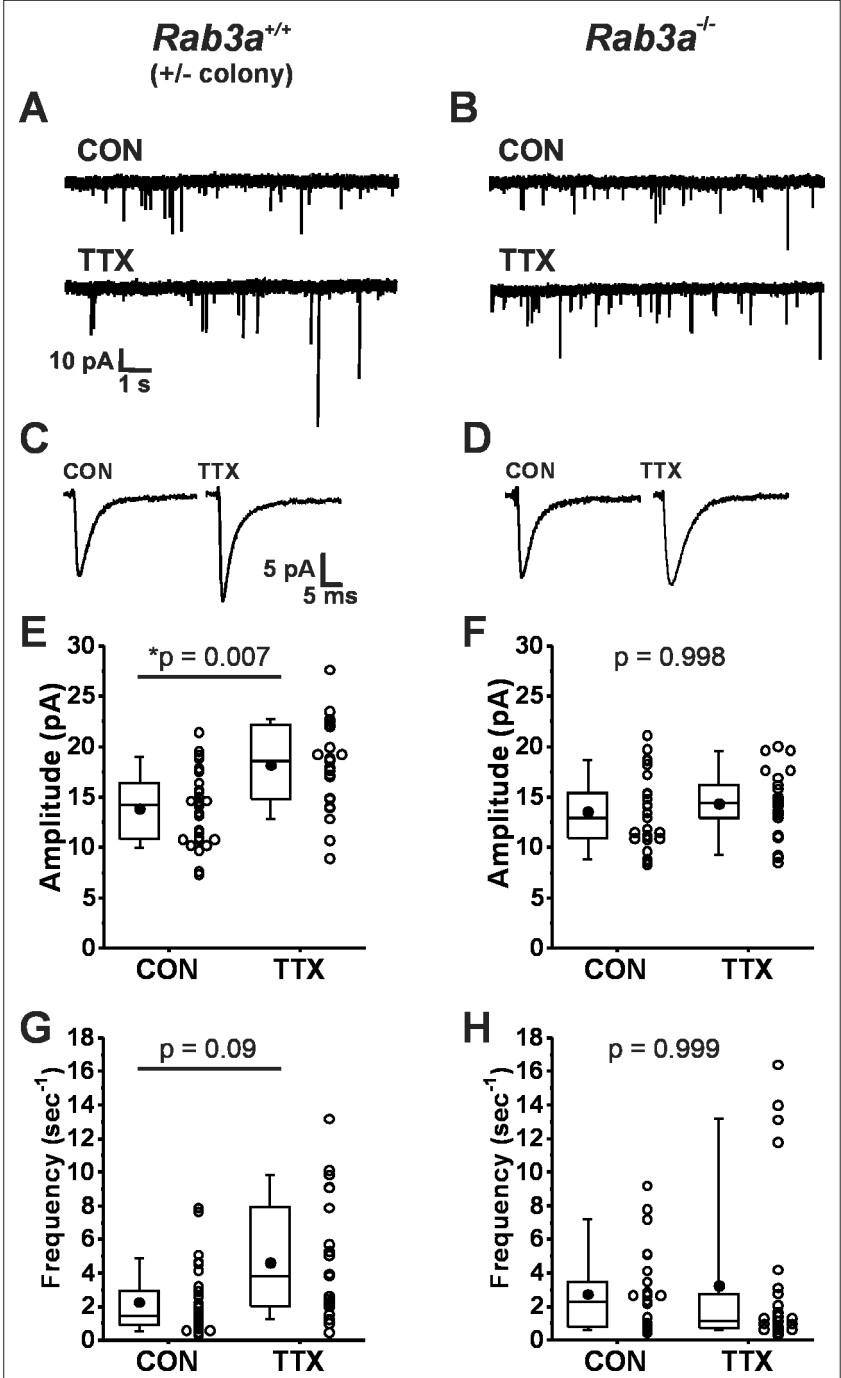

**Figure 1.** Loss of *Rab3a* disrupted the TTX-induced increase in amplitudes of miniature excitatory postsynaptic currents (mEPSCs) recorded in cultured mouse cortical neurons; the increase in frequency was more variable but also appears to be reduced. (**A**) Ten-second example traces recorded at −60 mV in pyramidal cortical neurons from an untreated (CON) and TTX-treated (TTX) neuron in cultures prepared from *Rab3a*+/+ mice from the *Rab3a*+/− colony. (**B**) Ten-second example traces recorded at −60 mV in pyramidal cortical neurons from an untreated (CON) and TTX-treated (TTX) neuron in cultures prepared from *Rab3a*−/− mice. (**C, D**) Average traces for the recordings shown in A and B, respectively. (**E**) Box plots for average mEPSC amplitudes from untreated cells and TTX-treated cells in cultures prepared from *Rab3a*+/+ mice (*Rab3a*+/− colony; CON, *N* = 30 cells, 13.9 ± 0.7 pA; TTX, *N* = 23 cells, 18.2 ± 0.9 pA; from eleven cultures). (**F**) Box plots for average mEPSC amplitudes from untreated cells and TTX-treated cells in cultures prepared from *Rab3a*−/− mice (CON, *N* = 25 cells, 13.6 ± 0.7 pA; TTX, *N* = 26 cells, 14.3 ± 0.6 pA); from eleven cultures. (**G**) Box plots for average mEPSC frequency for same *Rab3a*+/+ cells as in (**E**) (CON,

*Figure 1 continued on next page*

*Figure 1 continued*

2.26 ± 0.37 s$^{-1}$; TTX, 4.62 ± 0.74 s$^{-1}$). (**H**) Box plots for average mEPSC frequency for same *Rab3a$^{-/-}$* cells as in (**F**) (CON, 2.74 ± 0.49 s$^{-1}$; TTX, 3.23 ± 0.93 s$^{-1}$). Box plot parameters: box ends, 25th and 75th percentiles; whiskers, 10th and 90th percentiles; open circles represent means from individual cells, bin size 0.1 pA, 0.5 s$^{-1}$; line, median; dot, mean. p-values (shown on the graphs) are from Tukey's post hoc test following a two-way ANOVA. For all p-values, * with underline indicates significance with p < 0.05.

The online version of this article includes the following source data for figure 1:

**Source data 1.** Source data used in *Figure 1* to show loss of TTX effect in Rab3a-/- mouse cortical neuron cultures.

(CON, *Rab3a$^{+/+}$*, 11.0 ± 0.7 pA, vs. CON, *Rab3a$^{Ebd/Ebd}$*, 15.1 ± 1.0 pA, p = 0.04, Tukey means comparison after two-way ANOVA). The increase in mEPSC amplitude in cultures from *Rab3a$^{Ebd/Ebd}$* mice is consistent with the increase in mEPC amplitude we observed at the *Rab3a$^{Ebd/Ebd}$* NMJ (*Wang et al., 2011*).

The small GTPase *Rab3a* is generally thought to function presynaptically to regulate synaptic vesicle trafficking, possibly in an activity-dependent manner (*Castillo et al., 1997*; *Lonart et al., 1998*; *Leenders et al., 2001*; *Schlüter et al., 2006*; *Coleman and Bykhovskaia, 2009*; *Tian et al., 2012*). In contrast, homeostatic plasticity of mEPSC amplitude has been attributed to an increase in postsynaptic receptors on the surface of the dendrite (*O'Brien et al., 1998*; *Turrigiano et al., 1998*). At the NMJ in vivo, we could find no evidence for an increase in AChRs after TTX block of the sciatic nerve in vivo (*Wang et al., 2005*). Is *Rab3a* required for the homeostatic increase in surface AMPA-type glutamate receptors that has been confirmed by multiple studies (see Introduction)?

In some studies of homeostatic synaptic plasticity, it has been found that application of a Ca$^{2+}$-permeable AMPA receptor-specific inhibitor reversed the homeostatic increase in mEPSC amplitude (*Ju et al., 2004*; *Thiagarajan et al., 2005*; *Sutton et al., 2006*). *Figure 3A* shows that the *TTX-induced increase* in mean mEPSC amplitude was nearly identical in a set of 11 CON and 11 TTX-treated cells before and after treatment with 1-naphthyl acetyl spermine trihydrochloride (NASPM), a synthetic analogue of the Ca$^{2+}$-permeable AMPA receptor inhibitor philanthotoxin (before NASPM, CON 12.9 ± 3.5 pA; TTX, 17.5 ± 3.1 pA; after NASPM, CON 11.9 ± 2.6 pA; TTX 16.1 ± 3.5 pA). This result indicates that the TTX-induced increase in mEPSC amplitude does not depend on Ca$^{2+}$-permeable receptors, since the effect of TTX clearly remained when their presence was removed by acute NASPM application. We do not think that there was any technical issue with the NASPM application, because overall, mEPSC amplitudes were reduced modestly in both untreated and TTX-treated cultures (*Figure 3B*, CON, before NASPM, 12.9 ± 3.5 pA; after NASPM, 11.9 ± 2.6 pA; TTX, before NASPM, 17.5 ± 3.1 pA; after NASPM, 16.1 ± 3.5 pA). Furthermore, NASPM consistently reduced mEPSC frequency (*Figure 3C*, CON, before NASPM, 1.84 ± 0.55 s$^{-1}$; after NASPM, 1.56 ± 0.53 s$^{-1}$; TTX, before NASPM, 4.40 ± 3.51 s$^{-1}$; after NASPM, 2.68 ± 2.25 s$^{-1}$). Similar to the data in *Figure 1*, frequency is significantly increased after TTX treatment (p = 0.04), but this difference is no longer significant after NASPM application (p = 0.20). One explanation for the effect on frequency is that there are synaptic sites that express only homomers of Ca$^{2+}$-permeable AMPA receptors (GluA2-lacking AMPARs); mEPSCs from these sites would be expected to be completely blocked by NASPM (see cartoon description in *Figure 3D*, left). We cannot rule out an alternative explanation, that presynaptic Ca$^{2+}$-permeable AMPA receptors normally enhance release probability, and NAPSM prevents this action. The magnitude of the frequency reduction following acute NASPM appears to be greater after TTX treatment, suggesting that loss of activity could promote the establishment of synaptic sites that contain only Ca$^{2+}$-permeable receptors. However, these new sites do not appear to contribute to the increase in mEPSC amplitude (*Figure 3A*). The lack of effect of NASPM rules out the contribution of homomeric Ca$^{2+}$-permeable AMPA receptors GluA1, GluA3, and GluA4 (*Hollmann et al., 1991*; *Burnashev et al., 1992*; *Jonas and Burnashev, 1995*), as well as Ca$^{2+}$-permeable forms of Kainate receptors GluA5 and GluA6 (*Koike et al., 1997*; *Sun et al., 2009*) to homeostatic synaptic plasticity.

Having established that homomeric Ca$^{2+}$-permeable receptors are not contributing to the homeostatic increase in mEPSC amplitude, we turned to immunofluorescence and confocal imaging to assess whether GluA2 receptor expression, which will identify GluA2 homomers and GluA2 heteromers (the former unlikely to contribute to mEPSCs given their low conductance relative to heteromers; *Swanson et al., 1997*; *Mansour et al., 2001*), was increased in our wild-type mouse cortical cultures following 48 hr treatment with TTX. Since mEPSCs necessarily report synaptic levels of receptors,

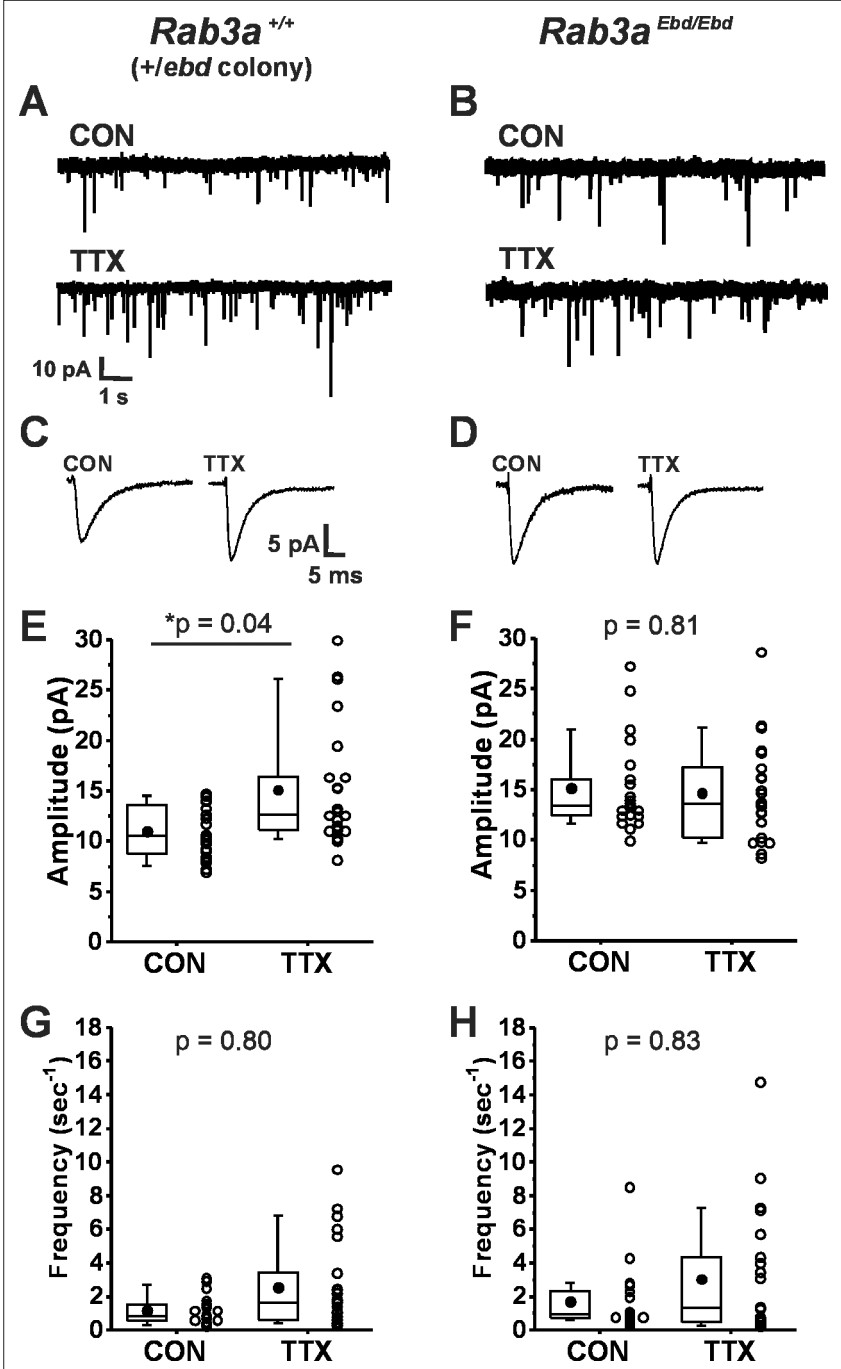

**Figure 2.** Normally functioning *Rab3a* was required for TTX-induced homeostatic plasticity of miniature excitatory postsynaptic current (mEPSC) amplitudes in cultured mouse cortical neurons. (**A**) Ten-second example traces recorded at −60 mV in pyramidal cortical neurons from an untreated (CON) and TTX-treated (TTX) neuron in cultures prepared from *Rab3a*⁺/⁺ mice from the *Rab3a*⁺/*Ebd* colony. (**B**) Ten-second example traces recorded at −60 mV in pyramidal cortical neurons from an untreated (CON) and TTX-treated (TTX) neuron in cultures prepared from *Rab3a*^Ebd/Ebd mice. (**C, D**) Average traces for the recordings shown in A and B, respectively. (**E**) Box plots for average mEPSC amplitudes from untreated cells and TTX-treated cells from cultures prepared from *Rab3a*⁺/⁺ mice (*Rab3a*⁺/*Ebd* colony; CON, $N$ = 20 cells, 11.0 ± 0.6 pA; TTX, $N$ = 23 cells, 15.1 ± 1.2 pA; from six cultures). (**F**) Box plots for average mEPSC amplitudes from untreated cells and TTX-treated cells from cultures prepared from *Rab3a*^Ebd/Ebd mice (CON, $N$ = 21 cells, 15.1 ± 1.0 pA; TTX, $N$ = 22 cells, 14.6 ± 1.1 pA; from seven cultures). (**G**) Box plots for average mEPSC frequency for same *Rab3a*⁺/⁺ cells as in (**E**) (CON, 1.15 ± 0.19 s⁻¹; TTX, 2.54 ± 0.55 s⁻¹). (**H**) Box plots for average mEPSC frequency for same *Rab3a*^Ebd/Ebd cells as in (**F**) (CON, 1.71 ± 0.41 s⁻¹; TTX, 3.05

*Figure 2 continued on next page*

*Figure 2 continued*

± 0.80 s⁻¹). Box plot parameters: box ends, 25th and 75th percentiles; whiskers, 10th and 90th percentiles; open circles represent means from individual cells; bin size 0.1 pA, 0.5 s⁻¹; line, median; dot, mean. p-values (shown on the graphs) are from a Tukey's post hoc test following a two-way ANOVA. For all p-values, * with underline indicates significance with p < 0.05.

The online version of this article includes the following source data for figure 2:

**Source data 1.** Source data for *Figure 2* showing loss of TTX effect on mEPSC amplitude in Rab3a Ebd/Ebd cortical neuron cultures.

we used VGLUT1 immunofluorescence to identify synapses on pyramidal primary apical dendrites labeled with MAP-2 immunofluorescence. We focused on the primary dendrite of pyramidal neurons as a way to reduce variability that might arise from being at widely ranging distances from the cell body or from inadvertently sampling dendritic regions arising from inhibitory neurons. In addition, it has been shown that there is a clear increase in response to glutamate in this region (*Turrigiano et al., 1998*). *Figure 4* (top) shows two pyramidal neurons, one from an untreated culture on the left, and one from a TTX-treated culture on the right, prepared from *Rab3a⁺/⁺* mice. To measure characteristics of synaptically located GluA2 receptor clusters, we zoomed in on the primary dendrite (region in white rectangle). The zoomed regions for single confocal sections of the selected area are shown below and contain pairs of VGLUT1- and GluA2-immunofluorescent clusters indicated by white trapezoidal frames. A dendrite typically required ~10 confocal sections to be fully captured, and the total number of synaptic pairs for all the sections imaged on a dendrite averaged 20 (20.4 ± 6.5 (SD); range, 11–38) so this is an atypically high number of pairs within a single section; these particular dendrites and sections were selected for illustration purposes. In addition to the synaptic pairs, we observed many GluA2-immunofluorescent clusters not associated with VGLUT1 immunofluorescence (a few are indicated with white arrows), most probably extrasynaptic receptors. There were also GluA2-positive clusters present outside of MAP2-positive dendrites, which may be located on astrocytes (*Fan et al., 1999*). Finally, GluA2-immunofluorescent clusters close to VGLUT1 immunofluorescence but not located along any apparent MAP2-positive neurites suggest the presence of axon–axonal contacts, although VGLUT1 has also been detected in astrocytes (*Ormel et al., 2012*). Only sites that contained both VGLUT1 and GluA2 immunofluorescence close to the primary MAP2-positive dendrite or a secondary branch along this primary dendrite were selected for analyses. In the experiment from which these images were collected, we analyzed the distance from the cell body for each synapse. The average distance from the cell body, for dendrites from the untreated cell, was 38.5 ± 2.8 µm; for the TTX-treated cell, it was 42.4 ± 3.2 µm (p = 0.35, Kruskal–Wallis test). Since the greatest distance from the cell body is set by the outer limit of the zoom window, and that window was placed adjacent to the cell body, distances in other experiments would be in this same range.

Variability in the magnitude of the homeostatic response from culture to culture is averaged out in physiological experiments by the pooling of data from many cells across many cultures. To reduce the necessity for many cultures, we chose to pair experiments in the same cultures by recording mEPSCs from one set of coverslips and processing another set of coverslips from the same culture for immunofluorescence. We also wanted to determine if the homeostatic response of mEPSC amplitudes varied from culture to culture, and if so, whether the GluA2 receptor levels varied in parallel. We completed this matched paradigm of physiology and immunofluorescence for three cultures prepared from *Rab3a⁺/⁺* mice. We first present the results as means for each culture, for control and TTX-treated dishes, for mEPSC data (*Figure 5A*) and imaging data (*Figure 5B*). Levels of GluA2 immunofluorescence at synaptic sites were quantified by the size of the GluA2-immunofluorescent receptor cluster, because this showed the greatest response to activity blockade; the average intensity value, and the integral of intensity values across the area of the cluster, are included in a summary of the data in *Table 1*. Three of three cortical cultures showed an increase in mean mEPSC amplitude after TTX treatment. In contrast, one of the same three cultures showed a decrease in mean GluA2 receptor cluster size, suggesting that the decrease in receptor levels in Culture #3 was not due to a lack of homeostatic effect on mEPSC amplitudes in that culture. Because there was a clear trend toward increased GluA2 receptor cluster size, but the receptor response appears more variable, we performed immunofluorescence measurements on two additional cortical cultures, both of which showed modest increases after TTX treatment (*Figure 5C*). We confirmed that there was also not a consistent increase in GluA1

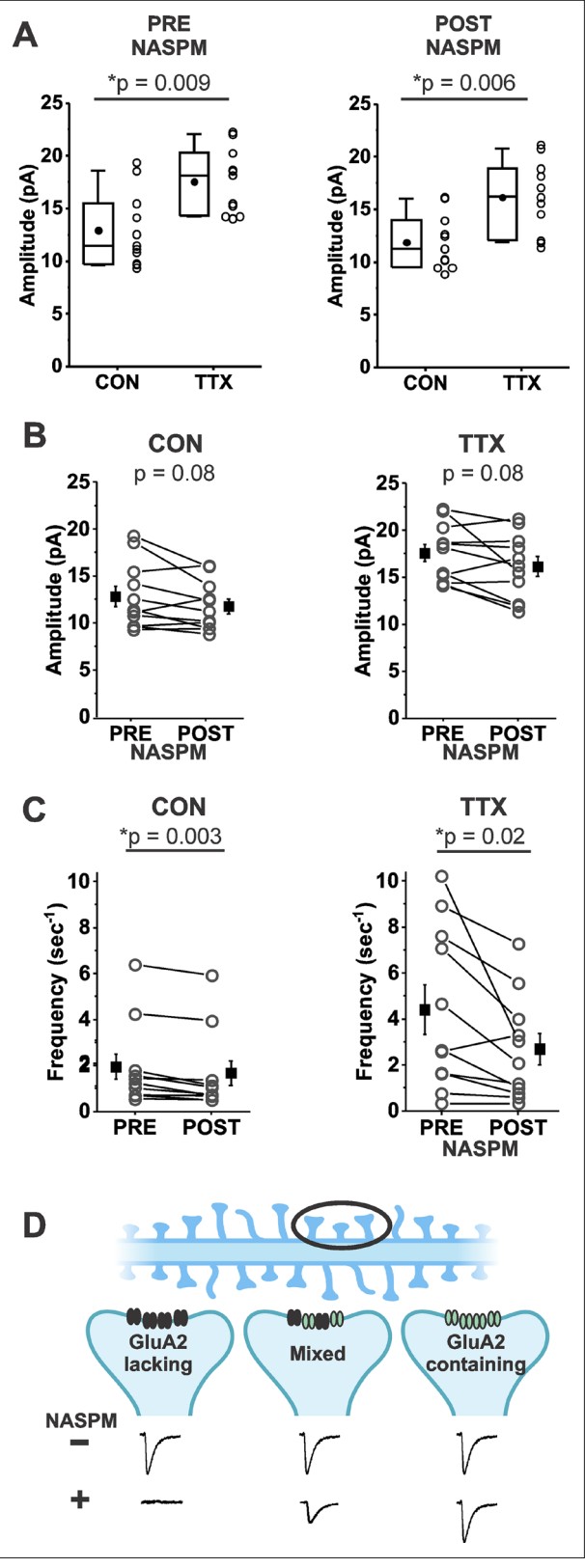

**Figure 3.** Homeostatic plasticity of miniature excitatory postsynaptic current (mEPSC) amplitudes in mouse cortical cultures treated with TTX for 48 hr was unchanged by acute inhibition of $Ca^{2+}$-permeable AMPA receptors by NASPM (20 μm). (**A**) Box plot comparison of the TTX effect on mEPSC amplitudes in the same pyramidal neurons before and after application of 20 μM NASPM ($N$ = 11 cells from three cultures; Pre-NASPM, CON: 12.9

*Figure 3 continued on next page*

*Figure 3 continued*

± 1.1 pA; TTX: 17.5 ± 0.9 pA; post-NASPM, CON: 11.9 ± 0.8 pA; TTX: 16.1 ± 1.0 pA). (**B**) Line series plot of mEPSC amplitudes before and after acute perfusion with 20 µM NASPM for untreated and TTX-treated pyramidal neurons; same cells as in (**A**); CON, pre-NASPM: 12.9 ± 1.1 pA; post-NASPM: 11.9 ± 0.8 pA; TTX, pre-NASPM: 17.5 ± 0.9 pA; post-NASPM: 16.1 ± 0.1 pA. (**C**) Line series plot of mEPSC frequency before and after acute perfusion with 20 µM NASPM for untreated and TTX-treated pyramidal neurons; same cells as in (**A**); CON, pre-NASPM: 1.84 ± 0.55 s$^{-1}$; post-NASPM; 1.56 ± 0.53 s$^{-1}$; TTX, pre-NASPM: 4.40 ± 1.06 s$^{-1}$; post-NASPM, 2.68 ± 0.68 s$^{-1}$. (**D**) A proposed mechanism for why NASPM had a robust effect on frequency without greatly affecting amplitude. A dendrite with spines (top) is expanded on three postsynaptic sites (middle) to show possible types of AMPA receptor distributions: left, at a site comprised only of Ca$^{2+}$-permeable AMPA receptors (NASPM-sensitive, GluA2-lacking receptors (black)), NASPM would completely inhibit the mEPSC, causing a decrease in frequency to be measured in the overall population; right, at a site comprised only of Ca$^{2+}$-impermeable AMPA receptors (NASPM-insensitive, GluA2-containing receptors (green)), NASPM would have no effect on mEPSC amplitude; middle, at a site comprised of a mix of Ca$^{2+}$-permeable and Ca$^{2+}$-impermeable AMPA receptors, NASPM would partially inhibit the mEPSC. Diagram was created using BioRender.com. Box plot parameters: box ends, 25th and 75th percentiles; whiskers, 10th and 90th percentiles; open circles represent means from individual cells; bin size 0.1 pA; line, median; dot, mean. p-values (shown on the graphs) are from Kruskal–Wallis test. Line series plot p-values are from Student's paired *t*-test. For all p-values, * with underline indicates significance with p < 0.05.

The online version of this article includes the following source data for figure 3:

**Source data 1.** Source data for effects of NASPM on untreated and TTX-treated cortical neuron cultures.

receptor levels in two matched cultures that showed robust homeostatic increases in mEPSC amplitudes (*Figure 5—figure supplement 1A*, culture mean mEPSC amplitudes; *Figure 5—figure supplement 1B*, culture mean GluA1 receptor cluster sizes).

We next pooled data from the multiple cultures to examine effects of activity blockade on GluA2 receptors at the level of cell means. Because the average number of synaptic sites identified in the primary dendrites in three cultures was 20.4, we also took 20 mEPSC samples from each cell, the 21st to the 40th mEPSC recorded, to minimize any differences in variability between GluA2 receptors and mEPSCs that might be due to a difference in sampling. In the data pooled from the three matched experiments, neurons from *Rab3a*$^{+/+}$ cultures showed a significant increase in mean mEPSC amplitudes following activity blockade (*Figure 5D*; *Table 1*). This result indicates that the homeostatic response averaged across the 3 cultures was very similar to, but slightly smaller than, that of the dataset presented in *Figure 1*. The mean for size of GluA2-immunofluorescent receptor clusters showed a trend toward higher values after activity blockade (18.1%) that was of similar magnitude to the mEPSC amplitude increase (19.7%), but did not reach statistical significance for three cultures (*Figure 5E*) or five cultures (*Figure 5F*; *Table 1*). As expected, mean mEPSC amplitude was not increased following activity blockade in the data pooled from a new set of three *Rab3a*$^{-/-}$ cultures (*Table 1*). For immunofluorescence results from the same *Rab3a*$^{-/-}$ cultures, no trend toward higher values was apparent for GluA2 receptor cluster characteristics (*Table 1*). However, given the variability of the GluA2 response in cultures from *Rab3a*$^{+/+}$ mice, it would require a much higher number of *Rab3a*$^{+/+}$ and *Rab3a*$^{-/-}$ cultures to make any firm conclusions about the role of *Rab3a* in the homeostatic response of GluA2 receptors.

Having used VGLUT1 immunofluorescence to mark synaptic sites, we could also ask whether there was a presynaptic effect of activity blockade on the size, intensity, or integral of the VGLUT1 signal at sites apposed to the previously examined GluA2 postsynaptic sites. In the pooled data from cultures prepared from *Rab3a*$^{+/+}$ mice, we found no change in the size of the VGLUT1-positive regions and trends toward reduced intensity and integral (*Table 2*). These data do not support the idea that activity blockade caused an increase in the amount of VGLUT1 per vesicle, although we cannot rule out that a simultaneous reduction in the number of vesicles obscured the effect, since we did not have an independent label for synaptic vesicles. In cultures prepared from *Rab3a*$^{-/-}$ mice, the size, intensity, and integral of the VGLUT1-positive regions did not significantly change (*Table 2*).

We have demonstrated here for the first time that the synaptic vesicle protein *Rab3a* influences the homeostatic regulation of quantal amplitude. What are possible ways that *Rab3a* could exert its effect? It has previously been shown that exogenous addition of Tnfa to hippocampal cultures causes an increase in surface expression of GluA1 receptors (although not GluA2 receptors), and that the homeostatic increase in mEPSC amplitudes is abolished in cultures prepared from the *Tnfa* deletion

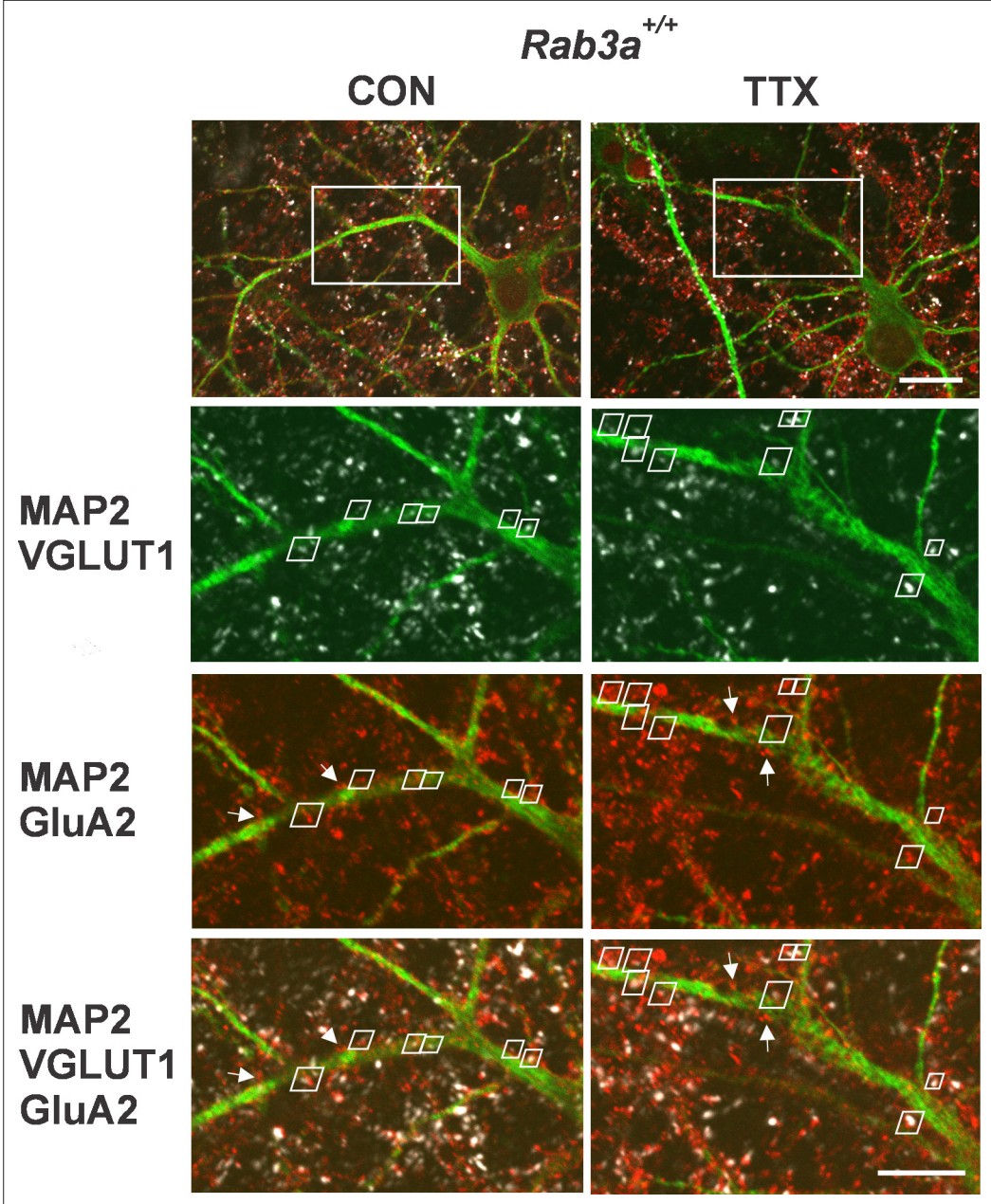

**Figure 4.** Identification of synaptic GluA2 receptor immunofluorescence on primary dendrites of pyramidal neurons in high-density mouse cortical co-cultures prepared from *Rab3a*[+/+] mice. (Top) Non-zoomed single confocal sections collected with a 60X oil immersion objective of recognizably pyramidal-shaped neurons, presumed to be excitatory neurons, selected for synaptic GluA2 analysis. A neuron was selected from an untreated coverslip (CON, left) and a TTX-treated coverslip (TTX, right) from the same culture prepared from *Rab3a*[+/+] animals (=*Rab3a*[+/+] Culture #2 in *Figure 5*). The white rectangular boxes indicate 5X zoomed areas shown in the images below. Note that the depth of the single confocal section of the non-zoomed neuron image is not at the same depth as the confocal section in the zoomed dendritic image, so some features are not visible in both images. Scale bar, 20 µm. (Bottom) 5X zoom single confocal sections selected for demonstration purposes because they had an unusually high number of identified synaptic pairs along the primary dendrite contained within a single confocal section. Synaptic pairs, highlighted with white trapezoids, were identified based on close proximity of GluA2 (red) and VGLUT1 (white) immunofluorescence, apposed to the MAP2 immunofluorescent primary dendrite (green). Some apparent synaptic pairs are not highlighted with a white trapezoid because there was a different confocal section in which the two immunofluorescent sites were maximally bright. There were GluA2-positive clusters located on the primary dendrites that were not apposed to VGLUT1-positive terminals (four of

*Figure 4 continued on next page*

*Figure 4 continued*

these non-synaptic GluA2 clusters are highlighted with white arrows in the MAP2-GluA2 and MAP2-GluA2-VGLUT1 panels). Scale bar, 10 μm. Images have been enhanced for visualization purposes only. No image manipulation was performed prior to signal quantification.

mouse (*Stellwagen et al., 2005*; *Stellwagen and Malenka, 2006*). Furthermore, neurons from $Tnfa^{+/+}$ mice plated on glial feeder layers derived from $Tnfa^{-/-}$ mice fail to show the increase in mEPSC amplitude after TTX treatment, indicating that the *Tnfa* inducing the receptor increases following activity blockade comes from the glial cells (*Stellwagen and Malenka, 2006*). It has recently been shown that the source of *Tnfa* is astrocytes rather than microglia (*Heir et al., 2024*).

*Rab3a* has been detected in astrocytes (*Maienschein et al., 1999*; *Hong et al., 2016*), so to determine whether *Rab3a* is acting via regulating *Tnfa* release from astrocytes, we performed experiments similar to those of *Stellwagen and Malenka, 2006* (schema illustrated in *Figure 6*, left). Under our culturing conditions, the glial feeder layers would be expected to be composed mainly of astrocytes (*Heir et al., 2024*). We compared the effect of activity blockade on mEPSC amplitudes recorded from cortical neurons from $Rab3a^{+/+}$ mice plated on $Rab3a^{+/+}$ glial feeder layers (*Figure 6A*); neurons from $Rab3a^{+/+}$ mice plated on $Rab3a^{-/-}$ glia (*Figure 6B*), and neurons from $Rab3a^{-/-}$ mice plated on $Rab3a^{+/+}$ glia (*Figure 6C*). If *Rab3a* is required for Tnfa release from glia, then $Rab3a^{+/+}$ neurons plated on $Rab3a^{-/-}$ glia should not show a homeostatic increase in mEPSC amplitude after treatment with TTX, and any cultures with $Rab3a^{+/+}$ glia should have a normal homeostatic response. We found the opposite result: mEPSC amplitudes increased in cultures where *Rab3a* was present in neurons (WT on WT, CON, 13.3 ± 0.5 pA, TTX, 16.7 ± 1.2 pA; WT on KO, CON, 13.3 ± 1.0 pA, TTX, 18.8 ± 1.4 pA), but the increase was greatly diminished in cultures where *Rab3a* was present only in glia (KO on WT, CON, 15.2 ± 1.1 pA, TTX, 16.9 ± 0.7 pA). In the glial feeder layer culture, the loss of *Rab3a* in the neurons may have caused an increase in mEPSC amplitude at baseline, similar to the Earlybird mutant in dissociated cultures, but this difference was not significant (CON, WT on WT, 13.3 ± 0.5 pA; CON, KO on WT, 15.2 ± 1.1 pA, p = 0.78, Tukey's post hoc test after two-way ANOVA). Regarding mEPSC frequency, in cultures with $Rab3a^{+/+}$ neurons, frequency trended upward after activity blockade, whereas in the culture with $Rab3a^{-/-}$ neurons, frequency was already increased (although not significantly, CON, WT on WT, 2.54 ± 0.57 s$^{-1}$; CON, KO on WT, 4.46 ± 1.21 s$^{-1}$, p = 0.59, Tukey's), and trended downward after activity blockade.

In summary, in the absence of *Rab3a* from glia, the homeostatic plasticity of mEPSC amplitude following activity blockade was normal, whereas in the absence of *Rab3a* in neurons, homeostatic plasticity was greatly diminished. This result makes it highly unlikely that *Rab3a* is required for the release of Tnfa, or another factor from glia, that induces a homeostatic upregulation of postsynaptic receptors and thereby increases mEPSC amplitude following TTX treatment. Neuronal *Rab3a* appears to mediate the homeostatic increase in mEPSC amplitude following activity blockade.

## Discussion

We found that homeostatic synaptic plasticity of mEPSC amplitude in dissociated mixed cultures of mouse cortical neurons and glia behaved remarkably similar to the mouse NMJ in response to loss of *Rab3a* function: in cultures from $Rab3a^{-/-}$ mice, the increase in mEPSC amplitudes following prolonged network silencing by TTX was strongly diminished, and in cultures from $Rab3a^{Ebd/Ebd}$ mice, basal mEPSC amplitude was increased compared to that of wild-type cultures and was not further modified by TTX treatment. These results suggest that normal function of the synaptic vesicle protein *Rab3a* is required for the homeostatic increase of mEPSC amplitude in cortical cultures.

### Is *Rab3a* acting in the presynaptic cell to cause homeostatic increase in mEPSC amplitude?

The impetus for our current study was two previous studies in which we examined homeostatic regulation of quantal amplitude at the NMJ. An advantage of studying the NMJ is that synaptic ACh receptors are easily identified with fluorescently labeled α-bungarotoxin, which allows for very accurate quantification of postsynaptic receptor density. We were able to detect a known change due to mixing 2 colors of α-bungarotoxin to within 1% (*Wang et al., 2005*). Using this model synapse, we

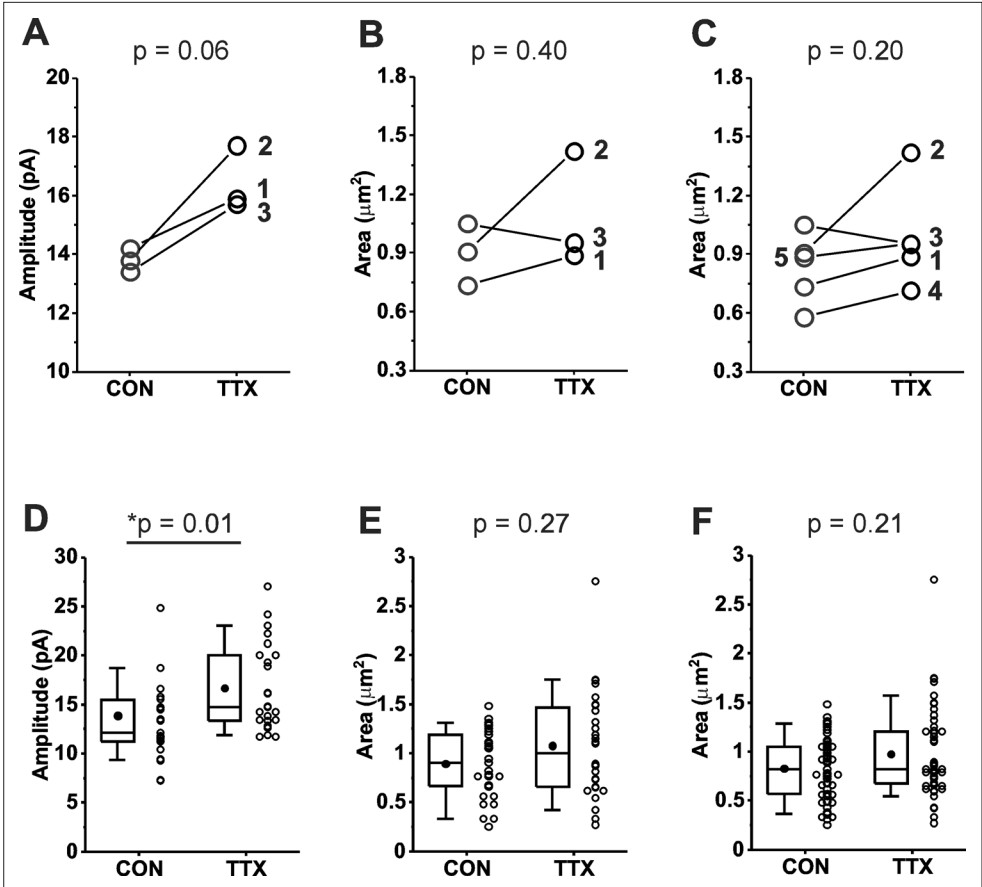

**Figure 5.** Comparison of miniature excitatory postsynaptic current (mEPSC) amplitudes and GluA2 receptor cluster areas in matched and unmatched mouse cortical cultures prepared from *Rab3a$^{+/+}$* mice and treated with TTX for 48 hr. (**A**) Culture averages of mEPSC amplitudes for untreated (CON) and TTX-treated coverslips (TTX) in each of three *Rab3a$^{+/+}$* mouse cortical co-cultures. Culture #1, CON, $N$ = 6, 14.2 ± 2.2 pA; TTX, $N$ = 6, 15.9 ± 1.9 pA; Culture #2, CON, $N$ = 7, 13.8 ± 2.4 pA; TTX, $N$ = 8, 17.7 ± 1.8 pA; Culture #3, CON, $N$ = 10, 13.4 ± 0.8 pA; TTX, $N$ = 9, 15.7 ± 1.0 pA. (**B**) Culture averages of GluA2 receptor cluster size in the same three cultures as shown in (**A**). Culture #1, CON, $N$ = 10, 0.73 ± 0.09 µm$^2$; TTX, $N$ = 9, 0.89 ± 0.09 µm$^2$; Culture #2, CON, $N$ = 9, 0.91 ± 0.12 µm$^2$; TTX, $N$ = 9, 1.42 ± 0.24 µm$^2$; Culture #3, CON, $N$ = 10, 1.05 ± 0.11 µm$^2$; TTX, $N$ = 9, 0.95 ± 0.15 µm$^2$. (**C**) Two additional culture averages are included that did not have corresponding mEPSC recordings. Culture #4, CON, $N$ = 10, 0.58 ± 0.07 µm$^2$; TTX, $N$ = 10, 0.71 ± 0.04 µm$^2$; Culture #5, CON, $N$ = 10, 0.89 ± 0.05 µm$^2$; TTX, $N$ = 10, 0.95 ± 0.08 µm$^2$. (**D**) Box plots of cell mean mEPSC amplitudes pooled from same three cultures as shown in (**A**). CON, $N$ = 23 cells, 13.7 ± 0.9 pA; TTX, $N$ = 24 cells, 16.4 ± 0.9 pA. (**E**) Box plots of dendrite mean GluA2 receptor cluster size pooled from same three cultures as shown in (**A**) and (**B**). CON, $N$ = 29 dendrites, 0.90 ± 0.06 µm$^2$; TTX, $N$ = 28 dendrites, 1.08 ± 0.10 µm$^2$. (**F**) Box plots of dendrite mean GluA2 receptor cluster size pooled from the same five cultures shown in (**C**). CON, $N$ = 49 dendrites, 0.83 ± 0.04 µm$^2$; TTX, $N$ = 48 dendrites, 0.98 ± 0.07 µm$^2$. Line series, Student's paired *t*-test; box plots, Kruskal–Wallis test. Box plot parameters: box ends, 25th and 75th percentiles; whiskers, 10 and 90 percentiles; open circles indicate means of individual cells/dendrites; bin size 0.1 pA, 0.05 µm$^2$; line, median; dot, mean. For all p-values, * with underline indicates significance with p < 0.05.

The online version of this article includes the following source data and figure supplement(s) for figure 5:

**Source data 1.** Source data for *Figure 5* comparing the TTX effect on mEPSC amplitudes and GluA2 receptors in matched and unmatched Rab3a+/+ cortical neuron cultures.

**Figure supplement 1.** Comparison of miniature excitatory postsynaptic current (mEPSC) amplitudes and GluA1 receptor cluster areas in matched mouse cortical cultures prepared from *Rab3a$^{+/+}$* mice and treated with TTX for 48 hr.

**Table 1.** Comparison of miniature excitatory postsynaptic current (mEPSC) amplitude and GluA2 receptor cluster characteristics in mouse cortical cultures prepared from *Rab3a*[+/+] mice and *Rab3a*[−/−] mice.
*p < 0.05, Kruskal–Wallis test.

| Quantity | *Rab3a*[+/+] | | | | *Rab3a*[−/−] | | | |
|---|---|---|---|---|---|---|---|---|
| | CON, mean, SEM | TTX, mean, SEM | % change | p-value | CON, mean, SEM | TTX, mean, SEM | % change | p-value |
| mEPSC amplitude | 13.7 ± 0.9 pA (23 cells, 3 cultures) | 16.4 ± 0.9 pA (24 cells, 3 cultures) | +19.7 | *0.02 | 14.9 ± 0.8 pA (21 cells, 3 cultures) | 13.5 ± 0.9 pA (21 cells, 3 cultures) | −9.4 | 0.18 |
| GluA2 cluster size | 0.83 ± 0.04 μm$^2$ (49 cells, 5 cultures) | 0.98 ± 0.07 μm$^2$ (48 cells, 5 cultures) | +18.1 | 0.21 | 0.93 ± 0.05 μm$^2$ (30 cells, 3 cultures) | 0.91 ± 0.05 μm$^2$ (29 cells, 3 cultures) | −2.2 | 0.75 |
| GluA2 cluster intensity | 733 ± 14 a.u. (49 cells, 5 cultures) | 738 ± 13 a.u. (48 cells, 5 cultures) | +0.7 | 0.69 | 766 ± 12 a.u. (30 cells, 3 cultures) | 776 ± 15 a.u. (29 cells, 3 cultures) | +1.3 | 0.46 |
| GluA2 cluster integral | 311,021 ± 19,282 a.u. (49 cells, 5 cultures) | 365,366 ± 27,080 a.u. (48 cells, 5 cultures) | +17.4 | 0.22 | 369,436 ± 23,439 a.u. (30 cells, 3 cultures) | 364,237 ± 25,833 a.u. (29 cells, 3 cultures) | −0.8 | 0.86 |

showed that there was no increase in synaptic AChRs after TTX treatment, whereas mEPC increased 35% (*Wang et al., 2005*). We further showed that the presynaptic protein *Rab3a* was necessary for full upregulation of mEPC amplitude (*Wang et al., 2011*). These data strongly suggested *Rab3a* contributed to homeostatic upregulation of quantal amplitude via a presynaptic mechanism. With the current study showing that *Rab3a* is required for the homeostatic increase in mEPSC amplitude in cortical neuron cultures, one interpretation is that in both situations, *Rab3a* is required for an increase in the presynaptic quantum.

The presynaptic quantum, or the amount of transmitter released during a single fusion event, is an important contributor to quantal size. The amount of transmitter released during vesicle fusion can be affected by the kinetics of the fusion pore opening (*Chang et al., 2017*). We previously used amperometric measurements in mouse adrenal chromaffin cells to show loss of *Rab3a* increased the occurrence of very small amplitude fusion pore feet (*Wang et al., 2008*). Paired with our finding that there is an increase in the occurrence of very slow-rising, abnormally shaped mEPCs at the NMJ of *Rab3a*[−/−] mice, our data suggest that small synaptic vesicles may have a fusion pore step under certain circumstances (see also *Chiang et al., 2018*). If activity blockade causes an increase in mEPSC amplitude due to a more rapid opening of a fusion pore, or a larger fusion pore conductance, it is possible that *Rab3a* is required for this modulation.

Another way to increase the presynaptic quantum is to increase levels of the transmitter transporter (*Daniels et al., 2004*; *De Gois et al., 2005*; *Wilson et al., 2005*; *Hartman et al., 2006*). However,

**Table 2.** Comparison of miniature excitatory postsynaptic current (mEPSC) amplitude and VGLUT1-positive presynaptic site characteristics in mouse cortical cultures prepared from *Rab3a*[+/+] and *Rab3a*[−/−] mice.
mEPSC data are identical to that in *Table 1*, reproduced here for comparison purposes. *p < 0.05, Kruskal–Wallis test.

| Quantity | *Rab3a*[+/+] | | | | *Rab3a*[−/−] | | | |
|---|---|---|---|---|---|---|---|---|
| | CON, mean, SEM | TTX, mean, SEM | % change | p-value | CON, mean, SEM | TTX, mean, SEM | % change | p-value |
| mEPSC amplitude | 13.9 ± 1.1 pA (23 cells, 3 cultures) | 16.7 ± 0.9 pA (24 cells, 3 cultures) | +20.1 | *0.01 | 14.9 ± 0.8 pA (21 cells, 3 cultures) | 13.5 ± 0.9 pA (21 cells, 3 cultures) | −9.4 | 0.18 |
| VGLUT1 site size | 1.17 ± 0.08 μm$^2$ (29 cells, 3 cultures) | 1.07 ± 0.06 μm$^2$ (24 cells, 3 cultures) | −8.5 | 0.97 | 0.89 ± 0.03 μm$^2$ (30 cells, 3 cultures) | 0.95 ± 0.04 μm$^2$ (29 cells, 3 cultures) | +6.7 | 0.55 |
| VGLUT1 site intensity | 699 ± 39 a.u. (29 cells, 3 cultures) | 639 ± 31 a.u. (28 cells, 3 cultures) | −8.6 | 0.28 | 554 ± 13 a.u. (30 cells, 3 cultures) | 570 ± 19 a.u. (29 cells, 3 cultures) | +2.9 | 0.18 |
| VGLUT1 site integral | 430,787 ± 36,818 a.u. (29 cells, 3 cultures) | 351,653 ± 21,689 a.u. (28 cells, 3 cultures) | −18.4 | 0.13 | 257,159 ± 18,265 a.u. (30 cells, 3 cultures) | 289,857 ± 18,521 a.u. (29 cells, 3 cultures) | +12.7 | 0.21 |

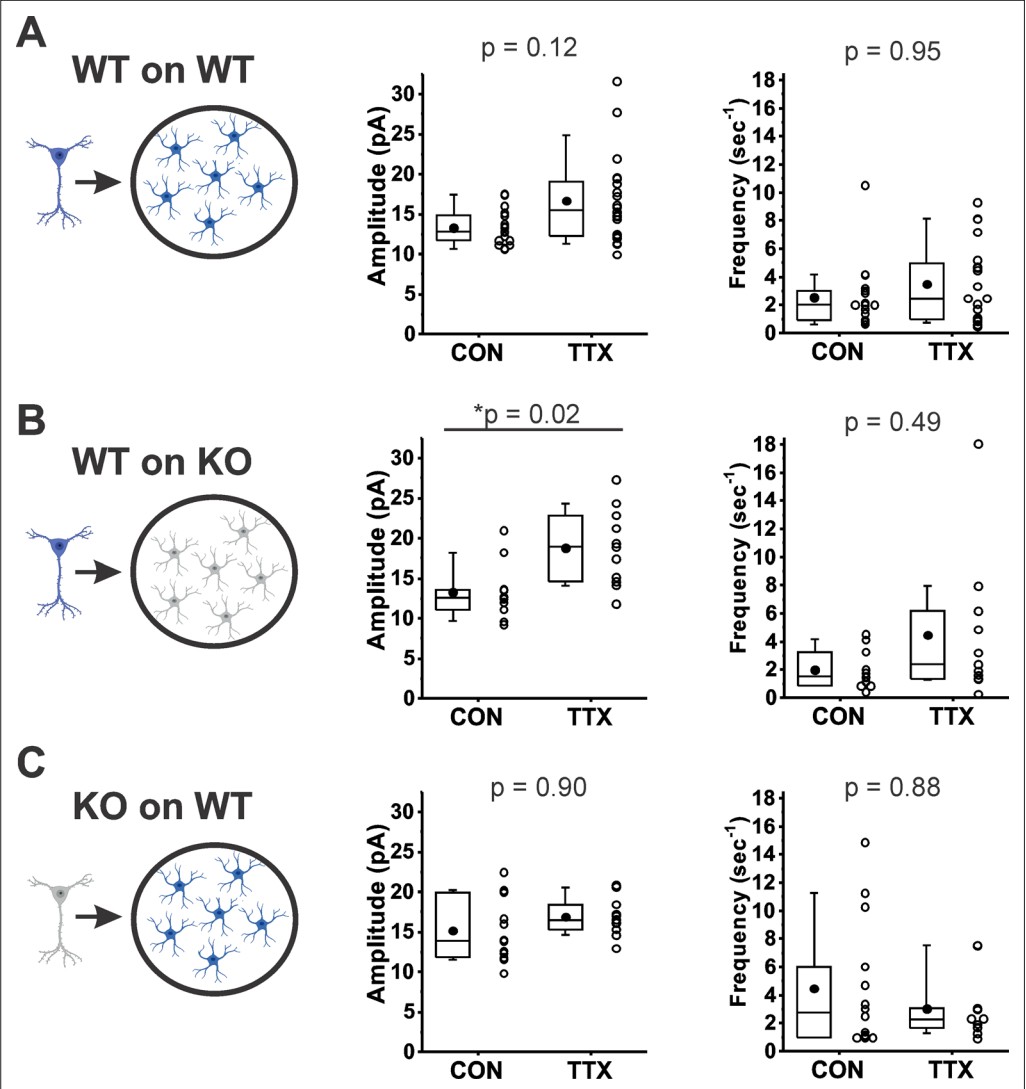

**Figure 6.** *Rab3a* in neurons, not astrocytes, was required for full TTX-induced homeostatic plasticity. (**A–C**) Miniature excitatory postsynaptic current (mEPSC) amplitude (middle) and frequency (right) data from dissociated cortical neurons plated on an astrocyte feeder layer, each prepared separately from the type of mice depicted in the schema (left): (**A**) Neurons from *Rab3a*[+/+] mice plated on astrocytes from *Rab3a*[+/+] mice. Box plots for mEPSC amplitudes (CON, *N* = 17 cells, 13.3 ± 0.5 pA; TTX, *N* = 20 cells, 16.7 ± 1.2 pA) and mEPSC frequency in the same recordings (mean, CON, 2.54 ± 0.57 s[−1]; TTX, 3.48 ± 0.64 s[−1]); from four cultures. (**B**) Neurons from *Rab3a*[+/+] mice plated on astrocytes from *Rab3a*[−/−] mice. Box plots for average mEPSC amplitude (CON, *N* = 11 cells, 13.3 ± 1.0 pA; TTX, *N* = 11 cells, 18.8 ± 1.4 pA) and average mEPSC frequency in the same recordings (CON, 2.01 ± 0.41 s[−1]; TTX, 4.47 ± 1.53 s[−1]); from two cultures. (**C**) Neurons from *Rab3a*[−/−] neurons plated on astrocytes from *Rab3a*[+/+] mice. Box plots for average mEPSC amplitude (CON, *N* = 14 cells, 15.2 ± 1.1 pA; TTX, *N* = 11 cells, 16.9 ± 0.7 pA) and mEPSC frequency in the same recordings (CON, 4.47 ± 1.21 s[−1]; TTX, 3.02 ± 0.70 s[−1]); from three cultures. Box plot parameters: box ends, 25th and 75th percentiles; whiskers, 10th and 90th percentiles, bin size 0.1 pA; open circles represent means from individual cells; line, median; dot, mean. p-values (shown on the graphs) are from Tukey's post hoc test following a two-way ANOVA. p-values denoted with * and underline indicate significance of $p < 0.05$.

we found that immunofluorescence for the glutamate transporter, VGLUT1, was not increased after activity blockade under our experimental conditions. A similar lack of increase in VGLUT1 levels, relative to synapsin, was previously observed in hippocampal cultures treated with NBQX, an AMPA receptor blocker (see *Figure 5—figure supplement 1*, *Wilson et al., 2005*). Taken together with our findings, it appears that there is not strong evidence that the amount of VGLUT1 at synapses is

a contributing factor to the increase in mEPSC amplitude in neuronal cultures after activity blockade with TTX.

Finally, the amount of transmitter released by a vesicle could be increased due to larger vesicle size. Larger vesicle diameter has been observed following activity disruption in multiple paradigms, including ocular deprivation, cortical ablation, and immobilization (*Kawana et al., 1971*; *Lloret and Saavedra, 1975*; *Cheresharov et al., 1978*). Loss of Rab3 family members is associated with increased vesicle size (*Riedel et al., 2002*; *Schonn et al., 2010*). An increase in volume has been shown to increase miniature endplate junctional current amplitude at the *Drosophila* NMJ (*Karunanithi et al., 2002*). *Rab3a* could be required for an activity-dependent increase in vesicle size, but the increase in diameter needed for a 25% increase in transmitter, and therefore mEPSC amplitude (~3%), would be difficult to detect.

## Is *Rab3a* acting on postsynaptic receptors to cause an increase in mEPSC amplitude?

In neurons, the increase in mEPSC amplitude following block of activity was not due to increases in $Ca^{2+}$-permeable GluA1 homomers because acute NASPM application was unable to reverse the TTX-induced increase in amplitude. While we found that the effect of TTX treatment on mEPSC amplitude and GluA2 receptor levels was of similar magnitude, the effect on GluA2 receptor levels was not statistically significant for the pooled datasets acquired in the same three cultures, nor in a larger dataset containing an additional two cultures. In light of the extensive literature documenting increases in GluA1 or GluA2 receptors, or both, after activity blockade, the less than convincing increase in GluA2 receptor levels in the current work came as a surprise. The reason for the lack of significance was that the increase in receptors was more variable. We are aware of three possible explanations for greater variability. (1) The number of synaptic sites sampled was smaller for the imaging dataset. This was because we limited our analysis to synaptic receptors on the primary dendrite, where a clear increase in sensitivity to exogenously applied glutamate was demonstrated (see Figure 3 in *Turrigiano et al., 1998*). To address this issue, we limited our sample size of mEPSCs in the matched experiments to the same number per cell as the mean sample size per cell for imaging, but there was still a discrepancy in the effect of TTX on mEPSCs versus receptors. (2) The small sample of only three cultures was not enough to reveal the increase in receptors. We attempted to address this by adding two more cultures. Although both additional cultures showed small increases in GluA2 receptor cluster size, when cells from these cultures were included in the pooled dataset, the p-value was not substantially improved. A remaining possibility is: (3) The complexity involved in processing cultures for immunofluorescence introduces variability that obscures the true effect. One can get around technical issues related to immunofluorescence processing by performing Western blots, but then any changes detected could be due to upregulation of extrasynaptic receptors.

What remains unclear is why our data were impacted by the variability inherent in immunofluorescence measurements, but data of other studies were not. Some other studies have used the number of synaptic sites or puncta as the sample size (e.g., *Hou et al., 2008*; *Ibata et al., 2008*; *Wang et al., 2019*). When we analyzed our data after including as few as 6 randomly selected cluster size measurements per cell, a Kruskal–Wallis test gave highly significant p-values for three cultures (p = 0.001, *n* = 174 sites (6 *29 cells)) and five cultures (p = 0.005, *n* = 294 sites (6 *49 cells)). We found only a single study that had comparable sample sizes for receptors and mEPSC amplitudes, and the p-value was <0.01 for mEPSC amplitudes (*n* = 9 and 10 cells for CON and TTX, respectively), but five times higher (<0.05) for receptor data (*n* = 13 and 14 cells for CON and TTX) (*Xu and Pozzo-Miller, 2017*). This single example is consistent with our suggestion that measurements of changes in receptors after activity blockade are more variable than measurements of mEPSC amplitudes. In sum, the lack of a robust increase in receptor levels leaves open the possibility that there is a presynaptic contributor to quantal size in mouse cortical cultures. However, the variability could arise from technical factors associated with the immunofluorescence method, and the mechanism of *Rab3a*-dependent homeostatic plasticity could be presynaptic for the NMJ and postsynaptic for cortical neurons.

Increased receptors likely contribute to increases in mEPSC amplitudes, but because we do not have a significant increase in GluA2 receptors in our experiments, it is impossible to conclude that the increase is lacking in cultures from *Rab3a*$^{-/-}$ neurons. Furthermore, without additional experiments targeting *Rab3a* deletion to the pre- or post-synaptic compartment, we cannot rule out that *Rab3a*

is regulating homeostatic plasticity in mouse cortical cultures by acting postsynaptically on receptor trafficking. To our knowledge, there is no evidence localizing *Rab3a* to postsynaptic dendritic sites, although other presynaptic molecules, such as SNARE proteins, synaptotagmins, and NSF, have been identified in the postsynaptic compartment (*Lledo et al., 1998*; *Nishimune et al., 1998*; *Osten et al., 1998*; *Song et al., 1998*; *Araki et al., 2010*; *Kennedy et al., 2010*; *Suh et al., 2010*; *Jurado et al., 2013*; *Hussain and Davanger, 2015*; *Gu et al., 2016*; *Wu et al., 2017*). A recent report implicates *Rab3a* in the localization of plasma membrane proteins, including the EGF receptor, in rafts in HEK cells and Jurkat-T cells, which indicates *Rab3a* can be found in other specialized locations besides the presynaptic nerve terminal (*Diaz-Rohrer et al., 2023*).

## Neuronal, not glial, *Rab3a* is required for homeostatic regulation of quantal amplitude

Astrocytic release of Tnfa was shown to mediate the increase in mEPSC amplitudes in prolonged TTX-treated cultures of dissociated hippocampal neurons and cultures of hippocampal slices (*Stellwagen and Malenka, 2006*; *Heir et al., 2024*). Tnfa accumulates after activity blockade, and directly applied to neuronal cultures, can cause an increase in GluA1 receptors, providing a potential mechanism by which activity blockade leads to the homeostatic upregulation of postsynaptic receptors (*Beattie et al., 2002*; *Stellwagen et al., 2005*; *Stellwagen and Malenka, 2006*). However, we found that loss of *Rab3a* in glia did not disrupt the increase in mEPSC amplitude after activity blockade, so it is unlikely that the *Rab3a*-dependent homeostatic increase in mEPSC amplitude is via the astrocytic-Tnfa pathway. Interestingly, we previously showed that the homeostatic increase in NMJ mEPC amplitude was completely normal in the absence of Tnfa (*Wang et al., 2011*), suggesting the possibility that neuronal *Rab3a* can act via a non-Tnfa mechanism to contribute to homeostatic regulation of quantal amplitude, although we have not ruled out a neuronal *Rab3a*-mediated Tnfa pathway in cortical cultures.

## Is the lack of homeostatic plasticity in the *Earlybird* mutant due to occlusion?

We found that mEPSC amplitude was increased in untreated cultures prepared from $Rab3a^{Ebd/Ebd}$ mice, compared to mEPSC amplitude in cultures from wild-type mice. If there is a limit to how large an mEPSC can become, it is possible that the mutant *Rab3a* does not affect homeostatic plasticity, only occludes it. Similar increases in mEPSC amplitude at baseline, combined with a failure to increase further after activity blockade, were noted in cultures prepared from mECP2, AKAP, Homer1a, and Arc/Arg3.1 deletion mice (*Shepherd et al., 2006*; *Hu et al., 2010*; *Xu and Pozzo-Miller, 2017*; *Sanderson et al., 2018*) as well as in cultures treated with ubiquitin proteasome inhibitors (*Jakawich et al., 2010*) or microRNA 186-5p (*Silva et al., 2019*), suggesting the possibility of a generalized phenomenon in which increased mEPSC amplitude is induced when homeostatic regulation is disrupted. Since we did not observe such an increase in mEPSC amplitude at baseline in cultures from $Rab3a^{-/-}$ mice, it remains a possibility that the point mutant may bind to novel partners, causing activities that would not be either facilitated or inhibited by *Rab3a*. Still, it is a strong coincidence that the novel activity of the mutant Earlybird *Rab3a* would affect mEPSC amplitude, the same characteristic that is modulated by activity blockade in a *Rab3a*-dependent manner. Although we cannot rule out occlusion, an alternate interpretation is that the presence of the mutant protein mimics the condition of activity blockade.

## Rab3a is the first presynaptic function protein required for full homeostatic plasticity of mEPSC amplitude

The number of molecules shown to be required for homeostatic synaptic plasticity of quantal size in neuronal cultures may have surpassed ~20 (see *Koesters et al., 2024* for a recent summary), but fall into two main categories: molecules involved in glutamate receptor expression or trafficking, and signaling molecules. To our knowledge, this is the first evidence that a presynaptic protein is participating in this process. Homeostatic synaptic plasticity is becoming increasingly implicated in essential normal functions such as sleep (*Tononi and Cirelli, 2014*; *Diering et al., 2017*; *Torrado Pacheco et al., 2021*), and pathological conditions such as epilepsy, neuropsychiatric disorders, Huntington's, and alcohol use disorder (*Trasande and Ramirez, 2007*; *Fernandes and Carvalho,*

*2016*; *Wang et al., 2017*; *Lovinger and Abrahao, 2018*; *Smith-Dijak et al., 2019*; *Lignani et al., 2020*; *Suzuki et al., 2021*; *Kavalali and Monteggia, 2023*). Therefore, our identification of a novel *Rab3a*-dependent regulatory pathway in homeostatic synaptic plasticity may have important therapeutic implications. Our findings also further cement the importance of *Rab3a* in activity-dependent modulation of synaptic strength.

# Materials and methods

**Key resources table**

| Reagent type (species) or resource | Designation | Source or reference | Identifiers | Additional information |
|---|---|---|---|---|
| Strain, strain background (include species and sex here) | Rab3A knockout strain, C57BL/6J; (*Mus musculus*, mixed sex) | The Jackson Laboratory | B6;129S-*Rab3a^{tm1Sud}*/J (strain #: 002443); RRID:IMSR_JAX:002443 | |
| Strain, strain background (include species and sex here) | Rab3A *earlybird* strain, C57BL/6J; C3HeJ (*Mus musculus*, mixed sex) | *Kapfhamer et al., 2002* | | |
| Genetic reagent (*Escherichia coli*) | Bsp1286I | New England Biolabs | Cat. #: R0120S | Restriction enzyme |
| Antibody | anti-GluA (mouse, extracellular, monoclonal: clone 6C4) | Millipore | Cat. #: MAB397 RRID:AB_2113875 | 1:40 |
| Antibody | anti-MAP2 (chicken, polyclonal) | Abcam | Cat. #: ab5392 RRID:AB_2138153 | 1:2500 |
| Antibody | anti-VGLUT1 (rabbit, polyclonal) | Synaptic Systems | Cat. #: 135 303 RRID:AB_887875 | 1:4000 |
| Sequence-based reagent | D8Mit31-F | This paper | Rab3A PCR Primers (forward) | TCC TGT GAC CTC CAA CTG TG |
| Sequence-based reagent | D8Mit31-R | This paper | Rab3A PCR Primers (reverse) | GGC CCA AAA CTG AGC AAC |
| Sequence-based reagent | RabF1 | This paper | Rab3A *ebd* PCR Primers (forward) | TGA CTC CTT CAC TCC AGC CT |
| Sequence-based reagent | Dcaps3R | This paper | Rab3A *ebd* PCR Primers (reverse) | TGC ACT GCA TTA AAT GAC TCC T |
| Chemical compound, drug | Tetrodotoxin (TTX) | Tocris | Cat. #: 1069 | |
| Chemical compound, drug | *N*-naphthyl acetylspermine (NASPM) | Tocris | Cat. #: 2766 | |
| Chemical compound, drug | Picrotoxin | Sigma-Aldrich | P1675 | |
| Software, algorithm | MiniAnalysis | Synaptosoft | | |
| Other | Papain | Worthington Biochemical | LK003178 | Enzyme for neuronal cell dissociation |
| Other | Neurobasal-A | Gibco | 10888022 | Neuronal cell culture media |
| Other | B27 | Gibco | 17504044 | Serum-free culture media supplement |

## Animals

All animal procedures were performed in accordance with the policies of the Institutional Animal Care and Use Committee of Wright State University (Animal User Protocols: 789, 901, 982, and 1082 for mouse breedings, and 796, 906, and 1011 for neuronal cultures). To determine the role of *Rab3a* in homeostatic plasticity in mouse cortical cultures, we employed two distinct genetic mouse strains with altered *Rab3a* function. *Rab3a* deletion mice were generated as follows: *Rab3a*^{+/−} heterozygous mice descended from the *Rab3a*^{−/−} strain obtained from Jackson Laboratories B6;129S-*Rab3atm1Sud*/J (strain #: 002443) were bred and genotyped as previously described (*Kapfhamer et al., 2002*; *Wang et al., 2008*), and maintained as a colony using multiple heterozygous breeding pairs. Homozygous *Rab3a*^{+/+} or *Rab3a*^{−/−} mouse pups were obtained by breeding *Rab3a*^{+/+} pairs and *Rab3a*^{−/−} pairs, respectively, so that the pups were the first-generation progeny of a homozygous mating. This protocol

minimizes the tendency of multiple generations of *Rab3a*[+/+] and *Rab3a*[−/−] homozygous breedings to produce progeny that are more and more genetically distinct. *Rab3a*[Ebd/Ebd] mice were identified in an EU-mutagenesis screen of C57BL/6J mice, and after a cross to C3H/HeJ, were backcrossed for 3 generations to C57BL/6J (*Kapfhamer et al., 2002*). *Rab3a*[+/Ebd] heterozygous mice were bred at Wright State University and genotyped in a two-step procedure: (1) a PCR reaction (forward primer: TGA CTC CTT CAC TCC AGC CT; reverse primer: TGC ACT GCA TTA AAT GAC TCC T) followed by (2) a restriction digest with enzyme Bsp1286I (New England Biolabs) that distinguishes the Earlybird mutant by its different base-pair products. *Rab3a*[+/−] mice were backcrossed with *Rab3a*[+/+] mice from the Earlybird heterozygous colony for 11 generations in an attempt to establish a single wild-type strain, but differences in mEPSC amplitude and adrenal chromaffin cell calcium currents persisted, likely due to genes that are close to the *Rab3a* site, resulting in two wild-type strains: (1) *Rab3a*[+/+] from the *Rab3a*[+/−] colony, and (2) *Rab3a*[+/+] from the *Rab3a*[+/Ebd] colony.

## Primary culture of mouse cortical neurons

Primary dissociated cultures of mixed neuronal and glia populations were prepared as previously described (*Hanes et al., 2020*). Briefly, postnatal day 0–2 (P0–P2) *Rab3a*[+/+], *Rab3a*[−/−] or *Rab3a*[Ebd/Ebd] neonates were euthanized by rapid decapitation, as approved by the Wright State University Institutional Animal Care and Use Committee, and brains were quickly removed. Each culture was prepared from the cortices harvested from two animals; neonates were not sexed. Cortices were collected in chilled Neurobasal-A media (Gibco) with osmolarity adjusted to 270 mOsm and supplemented with 40 U/ml DNAse I (Thermo Fisher Scientific). The tissues were digested with papain (Worthington Biochemical) at 20 U/ml at 37°C for 20 min followed by trituration with a sterile, fire-polished Pasteur pipette, then filtered through a 100-μm cell strainer and centrifuged at 1100 rpm for 2 min. After discarding the supernatant, the pellet was resuspended in room temperature Neurobasal-A media (270 mOsm), supplemented with 5% fetal bovine serum (FBS) for glia growth, and 2% B-27 supplement to promote neuronal growth (Gibco), L-glutamine, and gentamicin (Thermo Fisher Scientific). Neurons were counted and plated at $0.15 \times 10^6$ cells/coverslip onto 12 mm coverslips pre-coated with poly-L-lysine (BioCoat, Corning). This plating density produces a 'high-density' culture characterized by a complex mesh of neuronal processes criss-crossing the field of view, completely filling the space between cell bodies (see *Figure 4*). We found in preliminary experiments that an increase in mEPSC amplitude in mouse cortical cultures was inconsistent when cell density was low, likely due to lower levels of baseline activity, although it is a limitation of this study that we did not directly measure activity levels in cultures prepared from *Rab3a*[+/+], *Rab3a*[−/−], or *Rab3a*[Ebd/Ebd] mice, and therefore do not know if differences in results are due to differences in activity levels. While unlikely for *Rab3a*[+/+] and *Rab3a*[−/−] studies, given the mEPSC amplitude in untreated cultures was relatively small, it remains possible that in *Rab3a*[Ebd/Ebd] studies, the larger mEPSC amplitude in untreated cultures was due to a loss of activity in these cultures. The culture media for the first day (0 DIV) was the same as the above Neurobasal-A media supplemented with FBS, B-27, L-glutamine, and gentamicin and was switched after 24 hr (1 DIV) to media consisting of Neurobasal-A (270 mOsm), 2% B-27, and L-glutamine without FBS to avoid its toxic effects on neuronal viability and health (*Stellwagen and Malenka, 2006*). Half of the media was changed twice weekly and experiments were performed at 13–14 DIV. Two days prior to experiments, tetrodotoxin (TTX) (500 nM; Tocris), a potent Na[+] channel blocker, was added to some cultures to chronically silence all network activity and induce homeostatic synaptic plasticity mechanisms, while untreated sister cultures served as controls. Cultures prepared from mutant mice were compared with cultures from wild-type mice from their respective colonies. Note that the cultures comprising the *Rab3a*[+/+] data here are a subset of the data previously published in *Hanes et al., 2020*. This smaller dataset was restricted to the time period over which cultures were also prepared from *Rab3a*[−/−] mice.

## Preparation of glial feeder layers

Glial feeder layers were prepared from the cortices of P0–P2 *Rab3a*[+/+] or *Rab3a*[−/−] mouse pups as described previously (*Stellwagen and Malenka, 2006*). Briefly, cortices were dissected and cells were dissociated as described above. Cell suspensions of mixed neuronal and glial populations were plated onto glass coverslips pre-coated with poly-L-lysine in Dulbecco's modified Eagle media (Thermo Fisher Scientific) supplemented with 5% FBS (to promote glial proliferation and to kill neurons), L-glutamine,

and gentamicin, and maintained in an incubator at 37°C, 5% $CO_2$; cultures were maintained in this manner for up to 1 month to generate purely glial cultures (all neurons typically died off by 7 DIV). Culture media was replaced after 24 hr, and subsequent media changes were made twice weekly, replacing half of the culture media with fresh media. Feeder layers were not used for neuronal seeding until all native neurons were gone and glia approached 100% confluency (visually inspected).

## Plating of neurons on glial feeder layers

Cortical neurons were obtained as described above. The cell pellet obtained was resuspended in Neurobasal-A (osmolarity adjusted to 270 mOsm) containing B27 (2%, to promote neuronal growth), L-glutamine, and 5-fluorodeoxyuridine (FdU, a mitotic inhibitor; Sigma). Addition of FdU was used to prevent glial proliferation and contamination of the feeder layer with new glia, promoting only neuronal growth on the feeder layers (FdU-containing media was used for the maintenance of these cultures and all subsequent media changes). Glial culture media was removed from the feeder layer cultures, and the neuronal cell suspension was plated onto the glial feeder cultures. The culture strategy used to distinguish the relative roles of neuronal and glial *Rab3a* is outlined in *Figure 6*, Left. At 1 DIV, all of the culture media was removed and replaced with fresh Neurobasal-A media containing FdU described above, and half of the media was replaced twice per week for all subsequent media changes. Cultures were maintained in a 37°C, 5% $CO_2$ incubator for 13–14 DIV.

## Whole-cell voltage clamp to record mEPSCs

At 13–14 DIV, mEPSCs from TTX-treated and untreated sister cultures of $Rab3a^{+/+}$ or $Rab3a^{-/-}$ neurons from the $Rab3a^{+/-}$ colony, or $Rab3a^{+/+}$ or $Rab3a^{Ebd/Ebd}$ neurons from the $Rab3a^{+/Ebd}$ colony, were recorded via whole-cell voltage clamp to assess the role of *Rab3a* in homeostatic synaptic plasticity. Recordings were taken from pyramidal neurons, which were identified visually by a prominent apical dendrite; images were taken of all cells recorded from. Cells were continuously perfused with a solution consisting of (in mM): NaCl (115), KCl (5), $CaCl_2$ (2.5), $MgCl_2$ (1.3), dextrose (23), sucrose (26), HEPES (4.2), pH = 7.2 (*Stellwagen and Malenka, 2006*). On the day of recording, the osmolarity of the media from the cultures was measured (normally 285–295 mOsm) and the perfusate osmolarity was adjusted to match the culture osmolarity to protect against osmotic shock to the neurons. To isolate glutamatergic mEPSCs, TTX (500 nM) and picrotoxin (50 µM) were included in the perfusion solution to block action potentials and GABAergic currents, respectively. The NMDA receptor antagonist, APV, was not included in the perfusion solution because in pilot studies, all mEPSCs were blocked by 10 µM CNQX and 50 µm picrotoxin, demonstrating no APV-sensitive mEPSCs were present (data not shown). Previous studies have described NMDA mEPSCs in neuronal cultures, but recordings were performed in 0 $Mg^{2+}$ (*Watt et al., 2000*; *Sutton et al., 2006*). In our recordings, the presence of extracellular $Mg^{2+}$ (1.3 mM) and TTX would mean that the majority of NMDA receptors are blocked by extracellular $Mg^{2+}$ at resting Vm. Patch electrodes (3.5–5 MΩ) were filled with an internal solution containing (in mM): K-gluconate (130), NaCl (10), EGTA (1), $CaCl_2$ (0.132), $MgCl_2$ (2), HEPES (10), pH = 7.2. Osmolarity was adjusted to 10 mOsm less than the perfusion solution osmolarity, which we have found improves success rate of achieving low access resistance recordings. Neurons were clamped at a voltage of −60 mV using an Axopatch 200B patch-clamp (Axon Instruments), recorded from for 2–5 min, and data were collected with Clampex 10.0/10.2 (Axon Instruments).

## NASPM application

The antagonist of $Ca^{2+}$-permeable AMPA receptors, *N*-naphthyl acetylspermine (NASPM, 20 µM; Tocris), was applied during recordings in a subset of experiments. NASPM is a synthetic analog of Joro Spider Toxin (JSTX) (*Koike et al., 1997*), and chemicals in this family block $Ca^{2+}$-permeable glutamate receptors (*Blaschke et al., 1993*; *Herlitze et al., 1993*; *Iino et al., 1996*). The presence of the edited GluA2 subunit disrupts both the $Ca^{2+}$ permeability and the sensitivity to JSTX and related substances (*Blaschke et al., 1993*; *Bochet et al., 1994*; *Jonas and Burnashev, 1995*). Therefore, 20 µM NASPM is expected to completely block AMPA receptors containing GluA1, GluA3, or GluA4 subunits (*Hollmann et al., 1991*; *Burnashev et al., 1992*), or the Kainate receptors containing GluA5 or GluA6 (*Koike et al., 1997*; *Sun et al., 2009*), but be ineffective against any heteromer or homomer containing GluA2 subunits. Because the effect of the spider toxins and their analogs is use-dependent (*Herlitze et al., 1993*; *Iino et al., 1996*; *Koike et al., 1997*), NASPM was applied with a depolarizing

high K⁺ solution (25 mM KCl, 95 mM NaCl) which we expected to trigger release of glutamate and opening of glutamate-activated receptors, allowing entry of NASPM into the pore. Baseline recordings were performed for 2 min in our standard perfusate, followed by perfusion of NASPM + High K⁺ solution for 45 s, and then perfusion of a NASPM only solution for 5 min, after which recording was recommenced for 5 min (because we found in pilot experiments that frequency was reduced following NASPM application).

## Analysis of mEPSCs

mEPSCs were manually selected using Mini Analysis software (Synaptosoft), a standard program used for mEPSC event detection and analysis. Records were filtered at 2 kHz using a low-pass Butterworth filter prior to selection. The program detection threshold was set at 3 pA but the smallest mEPSC selected was 4.01 pA. A fully manual detection process means that the same criterion ('does this *look* like an mEPSC?') is applied to all events, not just the false positives or the false negatives, which prevents the bias from being primarily at one end or the other of the range of mEPSC amplitudes. It is important to note that when performing the MiniAnalysis process, the researcher did not know whether a record was from an untreated cell or a TTX-treated cell.

## Immunofluorescence, microscopy, and data analysis

Primary cultures of mouse cortical neurons were grown for 13–14 DIV. Antibodies to GluA2 (mouse ab against N-terminal, EMD Millipore) were added directly to live cultures at 1:40 dilution, and incubated at 37°C in a $CO_2$ incubator for 45 min. Cultures were rinsed three times with PBS/5% donkey serum before being fixed with 4% paraformaldehyde. After 3 rinses in PBS/5% donkey serum, cultures were incubated in CY3-labeled donkey-anti-mouse secondary antibodies for 1 hr at room temperature, rinsed in PBS/5% donkey serum, permeabilized with 0.2% saponin, and incubated in chick anti-MAP2 (1:2500, AbCAM) and rabbit anti-VGLUT1 (1:4000, Synaptic Systems) for 1 hr at room temperature in PBS/5% donkey serum. After rinsing with PBS/5% donkey serum, coverslips were incubated with 488-anti chick and CY5-anti rabbit secondary antibodies for 1 hr at room temperature, rinsed, blotted to remove excess liquid, and inverted on a drop of Vectashield (Vector Labs). Coverslips were sealed with nail polish and stored at 4°C for <1 week before imaging. All secondary antibodies were from Jackson Immunoresearch and were used at 1:225 dilution.

Coverslips were viewed on a Fluoview FV1000 laser scanning confocal microscope with a 60x oil immersion, 1.35 NA objective. Once a pyramidal neuron was identified and a single confocal section of the cell acquired (see top, *Figure 4*), Fluoview 2.1 software was used to zoom in on the primary dendrite (5X) and a series of confocal sections were taken every 0.5 μm. Images were analyzed offline with ImagePro 6 (Cybernetics). The composite image was used to locate synaptic sites containing both VGLUT1 and GluA2 immunoreactivity in close apposition to each other and to the primary dendrite or a secondary branch identified with MAP2 immunoreactivity. An area of interest (AOI) was manually drawn around the GluA2 cluster in the confocal section in which it was the brightest. The AOIs for a dendrite were saved in a single file; the AOI number and the confocal section it was associated with were noted for later retrieval. For quantification, AOIs were loaded, an individual AOI was called up on the appropriate section, and the count/measurement tool was used to apply a threshold. Multiple thresholds were used, 400–550, but we found that within a given experiment, the magnitude of the TTX effect was not altered by different thresholds; therefore, all data presented here use a threshold of 450. Pixels within the cluster that were above the threshold were automatically outlined, and size, average intensity, and integral of the outlined region saved to an Origin2020 (OriginLab) file for calculating mean values. For analyzing VGLUT1 sites, previous AOIs were examined, and if not directly over the VGLUT1 site, moved to encompass it (this happened when GluA2 and VGLUT1 were side by side rather than overlapping). The threshold was 200 for all VGLUT1 data except *Rab3a*⁺/⁺ Culture #1, where the threshold was 400. There was no threshold that worked for both that culture and the other cultures, with too much of the background regions being included for Culture #1 with a threshold of 200, and too little of the synaptic site area being included for the other cultures with a threshold of 400.

## Statistics

In our previous publication (*Hanes et al., 2020*), we described the characteristics of homeostatic plasticity of mEPSC amplitudes in mouse cortical cultures using multiple mEPSC amplitude quantiles from each cell to create datasets with thousands of samples. We compared the cumulative distribution functions for mEPSC amplitudes in untreated versus TTX-treated cultures, as well as sorting the values from smallest to largest, and plotted TTX values versus control values in the rank order plot first described in *Turrigiano et al., 1998* to demonstrate synaptic scaling. Finally, we plotted the ratio of TTX values/control values versus control values to show that the ratio, which represents the effect of TTX, increased with increasing control values, which we termed 'divergent scaling'. Here, we wanted to avoid the inflation of sample size caused by pooling multiple mEPSCs per cell. To statistically compare the effects of TTX in cultures of wild-type mice, $Rab3a^{-/-}$ mice, and $Rab3a^{Ebd/Ebd}$ mutant mice, or, WT–WT, WT–KO, and KO–WT neuron–glial cultures, we performed a two-way ANOVA, followed by a Tukey test, and used the p-value for each specific situation, for example, CON versus TTX for $Rab3a^{+/+}$; CON versus TTX for $Rab3a^{-/-}$, and so on. The overall means are presented ± SEM in the text, and p-values are displayed above each plot, with $N$ = the number of cells. For the characteristics of synaptic GluA2 receptor clusters and VGLUT1 sites in imaging experiments, $N$ = the number of dendrites (one dendrite was sampled for each cell), and statistical comparisons were made with the non-parametric Kruskal–Wallis test. Although we quantified three characteristics of the GluA2 clusters and VGLUT1 sites: size, average intensity, and integral, we have not corrected p-values for multiple comparisons. Means are presented as box plots + data, with the box at 25 and 75 percentiles, whiskers at 10 and 90 percentiles, a dot indicating the mean, and a line indicating the median. In *Figure 3*, mEPSC amplitudes were measured within the same cell before and after NASPM application, so a paired *t*-test was used to compare the pre- and post-NASPM mEPSC amplitudes. In *Figure 5A–C*, the mean mEPSC amplitude for untreated and TTX-treated coverslips in the same culture is compared with a paired *t*-test. Statistics were computed and plots were created using Origin2020 (OriginLab).

## Acknowledgements

The authors thank E Whitlock and H Ghouse for their contributions to confocal image analyses. This work was partially supported by NINDS P01 NS057228, Tim Cope, PI, KE, PI project 4.

## Additional information

### Funding

| Funder | Grant reference number | Author |
| --- | --- | --- |
| National Institute of Neurological Disorders and Stroke | P01NS057228 | Kathrin Engisch |

The funders had no role in study design, data collection, and interpretation, or the decision to submit the work for publication.

### Author contributions

Andrew G Koesters, Conceptualization, Data curation, Formal analysis, Investigation, Methodology, Writing – original draft, Writing – review and editing; Mark M Rich, Conceptualization, Formal analysis, Writing – original draft, Writing – review and editing; Kathrin Engisch, Conceptualization, Data curation, Formal analysis, Supervision, Funding acquisition, Methodology, Writing – original draft, Project administration, Writing – review and editing

### Author ORCIDs

Andrew G Koesters  https://orcid.org/0000-0003-3281-188X
Mark M Rich  https://orcid.org/0000-0002-6956-5531
Kathrin Engisch  https://orcid.org/0000-0002-1058-5343

## Ethics

All animal procedures were performed in accordance with the policies of the Institutional Animal Care and Use Committee of Wright State University (Animal User Protocols: 789, 901, 982, and 1082 for mouse breedings, and 796, 906, and 1011 for neuronal cultures).

Reviewer #1 (Public review): https://doi.org/10.7554/eLife.90261.5.sa1
Reviewer #2 (Public review): https://doi.org/10.7554/eLife.90261.5.sa2
Reviewer #3 (Public review): https://doi.org/10.7554/eLife.90261.5.sa3
Author response https://doi.org/10.7554/eLife.90261.5.sa4

# Additional files

## Supplementary files

MDAR checklist

## Data availability

All data generated or analyzed during this study are included in the manuscript and supporting files. Source data files have been provided for Figures 1–3 and 5.

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
