## [Editor Report · eLife Assessment]

This **valuable** study presents findings on the role of the small GTPase Rab3A in homeostatic plasticity. While the study provides **solid** evidence for a requirement of Rab3A in homeostatic up-scaling in cultured mouse neurons, it does not provide a model of how Rab3A is involved in homeostatic plasticity. The work will be of interest to researchers in the field of synaptic transmission and synaptic plasticity.

---

## [Referee Report · Reviewer #1 (Public review)]

Koesters and colleagues investigated the role of the small GTPase Rab3A in homeostatic scaling of miniature synaptic transmission in primary mouse cortical cultures using electrophysiology and immunohistochemistry. The major finding is that TTX incubation for 48 hours does not induce an increase in the amplitude of excitatory synaptic miniature events in neuronal cortical cultures derived from Rab3A KO and Rab3A Earlybird mutant mice. NASPM application had comparable effects on mEPSC amplitude in control and after TTX, implying that Ca2+-permeable glutamate receptors are unlikely modulated during synaptic scaling. Immunohistochemical analysis revealed no significant changes in GluA2 puncta size, intensity, and integral after TTX treatment in control and Rab3A KO cultures. Finally, they provide evidence that loss of Rab3A in neurons, but not astrocytes, blocks homeostatic scaling. Based on these data, the authors propose a model in which neuronal Rab3A is required for homeostatic scaling of synaptic transmission, potentially through GluA2-independent mechanisms.

The major finding - impaired homeostatic up-scaling after TTX treatment in Rab3A KO and Rab3 earlybird mutant neurons - is supported by data of high quality. However, the paper falls short of providing any evidence or direction regarding potential mechanisms. The data on GluA2 modulation after TTX incubation are likely statistically underpowered and do not allow drawing solid conclusions, such as GluA2-independent mechanisms of up-scaling.

The study should be of interest to the field because it implicates a presynaptic molecule in homeostatic scaling, which is generally thought to involve postsynaptic neurotransmitter receptor modulation. However, it remains unclear how Rab3A participates in homeostatic plasticity.

Major (remaining) point:

(1) The current version of the abstract only includes the results on GluA2 immunofluorescence and mEPSC amplitude modulation after TTX treatment in control cultures, and a requirement for Rab3A in neurons instead of astrocytes. The major findings, including the block of the mEPSC amplitude increase upon TTX treatment in Rab3KO/EB mutants, are not mentioned. The abstract should be revised so that it reflects all major findings, potentially at the expense of citing previous work by the authors.

---

## [Referee Report · Reviewer #2 (Public review)]

First, I would like to thank the authors for the response. I acknowledge that the authors show in previous studies that Rab3A acts from the presynaptic side at the NMJ, and that is, as the authors indicate, their impetus for the current study. However, mechanisms observed at a completely different type of synapses cannot be used as an argument for conclusions here. The authors also acknowledge that they should restrict their conclusions to the data in the current study, and they are merely proposing interpretations. Then perhaps they should restrict these interpretations to the discussion rather than make this claim in the abstract (lines 44-47). Here the authors ask whether Rab3A is involved in the homeostatic increase of postsynaptic AMPARs, am I understanding it correctly that their conclusion for this question is "increase in AMPAR levels in WT cultures is more variable than those in mEPSCs so that it is impossible to determine if Rab3A is involved at all"? If so, then this question has not been answered and should not be regarded as one of the main conclusions with the data presented here. It also remains unclear to me how this piece of inconclusive data serves the main objective of the study.

The authors state at the end that the current study is just an extension of their previous work, and therefore their interpretations here further support the idea that Rab3A is acting presynaptically. I would argue that it is the conclusive data, rather than interpretations that lack concrete evidence, that support ideas and models. I think that we would all agree that immunostaining measurements can be very variable. However, if the authors are determined to use this approach to answer one of their major questions, then perhaps one way to significantly strengthen their conclusions is to find ways to somewhat overcome this technical limitation.

Finally, I thank the authors for addressing other minor concerns of mine.

---

## [Referee Report · Reviewer #3 (Public review)]

This manuscript presents a number of interesting findings that have the potential to increase our understanding of the mechanism underlying homeostatic synaptic plasticity (HSP). The data broadly support that Rab3A plays a role in HSP, although the site and mechanism of action remain uncertain.

The authors clearly demonstrate the Rab3A plays a role in HSP at excitatory synapses, with substantially less plasticity occurring in the Rab3A KO neurons. There is also no apparent HSP in the Earlybird Rab3A mutation, although baseline synaptic strength is already elevated. In this context, it is unclear if the plasticity is absent, already induced by this mutation, or just occluded by a ceiling effect due the synapses already being strengthened. Occlusion may also occur in the mixed cultures, when Rab3A is missing from neurons but not astrocytes. The authors do appropriately discuss these options. The authors have solid data showing that Rab3A is unlikely to be active in astrocytes, Finally, they attempt to study the linkage between changes in synaptic strength and AMPA receptor trafficking during HSP, and conclude that trafficking may not be solely responsible for the changes in synaptic strength during HSP.

Strengths:

This work adds another player into the mechanisms underlying an important form of synaptic plasticity. The plasticity is likely only reduced, suggesting Rab3A is only partially required and perhaps multiple mechanisms contribute. The authors speculate about some possible novel mechanisms, including whether Rab3A is active pre-synaptically to regulate quantal amplitude.

As Rab3A is primarily known as a pre-synaptic molecule, this possibility is intriguing and novel for this system. However, it is based on the partial dissociation of AMPAR trafficking and synaptic response, and lacks strong support. On average, they saw similar magnitude of change in mEPSC amplitude and GluA2 cluster area and integral, but the GluA2 data was not significant due to higher variability. It is difficult to determine if this is due to biology or methodology - the imaging method involves assessing puncta pairs (GluA2/VGlut1) clearly associated with a MAP2 labeled dendrite. This is a small subset of synapses, with usually less than 20 synapses per neuron analyzed, which would be expected to be more variable than mEPSC recordings averaged across several hundred events. However, when they reduce the mEPSC number of events to similar numbers as the imaging, the mESPC amplitudes are still less variable than the imaging data. The reason for this remains unclear. The pool of sampled synapses is still different between the methods and recent data has shown that synapses have variable responses during HSP. Further, there could be variability in the subunit composition of newly inserted AMPARs, and only assessing GluA2 could mask this (see below). It is intriguing that pre-synaptic changes might contribute to HSP, especially given the likely localization of Rab3A. But it remains difficult to distinguish if the apparent difference in imaging and electrophysiology is a methodological issue rather than a biological one. Stronger data, especially positive data on changes in release, will be necessary to conclude that pre-synaptic factors are required for HSP, beyond the established changes in post-synaptic receptor trafficking. Specific deletion of Rab3A from pre-synaptic neurons would also be highly informative.

Other questions arise from the NASPM experiments, used to justify looking at GluA2 (and not GluA1) in the immunostaining. First, there is a strong frequency effect that is unclear in origin. One would expect NASPM to merely block some fraction of the post-synaptic current, and not affect pre-synaptic release or block whole synapses. But the change in frequency seems to argue (as the authors do) that some synapses only have CP-AMPARs, while the rest of the synapses have few or none. Another possibility is that there are pre-synaptic NASPM-sensitive receptors that influence release probability. Further, the amplitude data show a strong trend towards smaller amplitude following NASPM treatment (Fig 3B). The p value for both control and TTX neurons was 0.08 - it is very difficult to argue that there is no effect. And the decrease on average is larger in the TTX neurons, and some cells show a strong effect. It is possible there is some heterogeneity between neurons on whether GluA1/A2 heteromers or GluA1 homomers are added during HSP. This would impact the conclusions about the GluA2 imaging as compared to the mEPSC amplitude data.

To understand the role of Rab3A in HSP will require addressing two main issues:

(1) Is Rab3A acting pre-synaptically, post-synaptically or both? The authors provide good evidence that Rab3A is acting within neurons and not astrocytes. But where it is acting (pre or post) would aid substantially in understanding its role. The general view in the field has been that HSP is regulated post-synaptically via regulation of AMPAR trafficking, and considerable evidence supports this view. More concrete support for the authors suggestion of a pre-synaptic site of control would be helpful.

(2) Rab3A is also found at inhibitory synapses. It would be very informative to know if HSP at inhibitory synapses is similarly affected. This is particularly relevant as at inhibitory synapses, one expects a removal of GABARs or a decrease in GABA release (ie the opposite of whatever is happening at excitatory synapses). If both processes are regulated by Rab3A, this might suggest a role for this protein more upstream in the signaling; an effect only at excitatory synapses would argue for a more specific role just at those synapses.

Comments on revisions:

The section on TNF is a bit odd. The data on the astrocyte deletion of Rab3A only argues that Rab3A is unlikely to regulate TNF release. But it could easily be downstream of the neuronal TNF receptor. Without any data addressing the TNF response, it seems quite premature to argue that Rab3A is part of a TNF-independent pathway.

The section title (line 506-7) declaring Rab3A as the first presynaptic protein involved in HSP is also premature, as they don't know it is acting pre-synaptically.

---

## [Author Response]

The following is the authors’ response to the previous reviews

General Response to Reviewers:

We thank the Reviewers for their comments, which continue to substantially improve the quality and clarity of the manuscript, and therefore help us to strengthen its message while acknowledging alternative explanations.

All three reviewers raised the concern that we have not proven that Rab3A is acting on a presynaptic mechanism to increase mEPSC amplitude after TTX treatment of mouse cortical cultures. The reviewers’ main point is that we have not shown a lack of upregulation of postsynaptic receptors in mouse cortical cultures. We want to stress that we agree that postsynaptic receptors are upregulated after activity block in neuronal cultures. However, the reviewers are not acknowledging that we have previously presented strong evidence at the mammalian NMJ that there is no increase in AChR after activity blockade, and therefore the requirement for Rab3A in the homeostatic increase in quantal amplitude points to a presynaptic contribution. We agree that we should restrict our firmest conclusions to the data in the current study, but in the Discussion we are proposing interpretations. We have added the following new text:

“The impetus for our current study was two previous studies in which we examined homeostatic regulation of quantal amplitude at the NMJ. An advantage of studying the NMJ is that synaptic ACh receptors are easily identified with fluorescently labeled alpha-bungarotoxin, which allows for very accurate quantification of postsynaptic receptor density. We were able to detect a known change due to mixing 2 colors of alpha-BTX to within 1% (Wang et al., 2005). Using this model synapse, we showed that there was no increase in synaptic AChRs after TTX treatment, whereas miniature endplate current increased 35% (Wang et al., 2005). We further showed that the presynaptic protein Rab3A was necessary for full upregulation of mEPC amplitude (Wang et al., 2011). These data strongly suggested Rab3A contributed to homeostatic upregulation of quantal amplitude via a presynaptic mechanism. With the current study showing that Rab3A is required for the homeostatic increase in mEPSC amplitude in cortical cultures, one interpretation is that in both situations, Rab3A is required for an increase in the presynaptic quantum.”

The point we are making is that the current manuscript is an extension of that work and interpretation of our findings regarding the variability of upregulation of postsynaptic receptors in our mouse cortical cultures further supports the idea that there is a Rab3Adependent presynaptic contribution to homeostatic increases in quantal amplitude.

**Public Reviews:**

**Reviewer #1 (Public review):**
Koesters and colleagues investigated the role of the small GTPase Rab3A in homeostatic scaling of miniature synaptic transmission in primary mouse cortical cultures using electrophysiology and immunohistochemistry. The major finding is that TTX incubation for 48 hours does not induce an increase in the amplitude of excitatory synaptic miniature events in neuronal cortical cultures derived from Rab3A KO and Rab3A Earlybird mutant mice. NASPM application had comparable effects on mEPSC amplitude in control and after TTX, implying that Ca2+-permeable glutamate receptors are unlikely modulated during synaptic scaling. Immunohistochemical analysis revealed no significant changes in GluA2 puncta size, intensity, and integral after TTX treatment in control and Rab3A KO cultures. Finally, they provide evidence that loss of Rab3A in neurons, but not astrocytes, blocks homeostatic scaling. Based on these data, the authors propose a model in which neuronal Rab3A is required for homeostatic scaling of synaptic transmission, potentially through GluA2-independent mechanisms.The major finding - impaired homeostatic up-scaling after TTX treatment in Rab3A KO and Rab3 earlybird mutant neurons - is supported by data of high quality. However, the paper falls short of providing any evidence or direction regarding potential mechanisms. The data on GluA2 modulation after TTX incubation are likely statistically underpowered, and do not allow drawing solid conclusions, such as GluA2-independent mechanisms of up-scaling.The study should be of interest to the field because it implicates a presynaptic molecule in homeostatic scaling, which is generally thought to involve postsynaptic neurotransmitter receptor modulation. However, it remains unclear how Rab3A participates in homeostatic plasticity.Major (remaining) point:(1) Direct quantitative comparison between electrophysiology and GluA2 imaging data is complicated by many factors, such as different signal-to-noise ratios. Hence, comparing the variability of the increase in mini amplitude vs. GluA2 fluorescence area is not valid. Thus, I recommend removing the sentence "We found that the increase in postsynaptic AMPAR levels was more variable than that of mEPSC amplitudes, suggesting other factors may contribute to the homeostatic increase in synaptic strength." from the abstract.

We have not removed the statement, but altered it to soften the conclusion. It now reads, “We found that the increase in postsynaptic AMPAR levels in wild type cultures was more variable than that of mEPSC amplitudes, which might be explained by a presynaptic contribution, but we cannot rule out variability in the measurement.”.

Similarly, the data do not directly support the conclusion of GluA2-independent mechanisms of homeostatic scaling. Statements like "We conclude that these data support the idea that there is another contributor to the TTX- induced increase in quantal size." should be thus revised or removed.

This particular statement is in the previous response to reviewers only, we deleted the sentence that starts, “The simplest explanation Rab3A regulates a presynaptic contributor….”. and “Imaging of immunofluorescence more variable…”. We deleted “ our data suggest….consistently leads to an increase in mEPSC amplitude and sometimes leads to….” We added “…the lack of a robust increase in receptor levels leaves open the possibility that there is a presynaptic contributor to quantal size in mouse cortical cultures. However, the variability could arise from technical factors associated with the immunofluorescence method, and the mechanism of Rab3A-dependent plasticity could be presynaptic for the NMJ and postsynaptic for cortical neurons.”

**Reviewer #2 (Public review):**
I thank the authors for their efforts in the revision. In general, I believe the main conclusion that Rab3A is required for TTX-induced homeostatic synaptic plasticity is wellsupported by the data presented, and this is an important addition to the repertoire of molecular players involved in homeostatic compensations. I also acknowledge that the authors are more cautious in making conclusions based on the current evidence, and the structure and logic have been much improved.The only major concern I have still falls on the interpretation of the mismatch between GluA2 cluster size and mEPSC amplitude. The authors argue that they are only trying to say that changes in the cluster size are more variable than those in the mEPSC amplitude, and they provide multiple explanations for this mismatch. It seems incongruous to state that the simplest explanation is a presynaptic factor when you have all these alternative factors that very likely have contributed to the results. Further, the authors speculate in the discussion that Rab3A does not regulate postsynaptic GluA2 but instead regulates a presynaptic contributor. Do the authors mean that, in their model, the mEPSC amplitude increases can be attributed to two factors- postsynaptic GluA2 regulation and a presynaptic contribution (which is regulated by Rab3A)? If so, and Rab3A does not affect GluA2 whatsoever, shouldn't we see GluA2 increase even in the absence of Rab3A? The data in Table 1 seems to indicate otherwise.

The main body of this comment is addressed in the General Response to Reviewers. In addition, we deleted text “current data, coupled with our previous findings at the mouse neuromuscular junction, support the idea that there are additional sources contributing to the homeostatic increase in quantal size.” We added new text, so the sentence now reads: “Increased receptors likely contribute to increases in mESPC amplitudes in mouse cortical cultures, but because we do not have a significant increase in GluA2 receptors in our experiments, it is impossible to conclude that the increase is lacking in cultures from Rab3A^-/-^ neurons.”

I also question the way the data are presented in Figure 5. The authors first compare 3 cultures and then 5 cultures altogether, if these experiments are all aimed to answer the same research question, then they should be pooled together. Interestingly, the additional two cultures both show increases in GluA2 clusters, which makes the decrease in culture #3 even more perplexing, for which the authors comment in line 261 that this is due to other factors. Shouldn't this be an indicator that something unusual has happened in this culture?Data in this figure is sufficient to support that GluA2 increases are variable across cultures, which hardly adds anything new to the paper or to the field.

A major goal of performing the immunofluorescence measurements in the same cultures for which we had electrophysiological results was to address the common impression that the homeostatic effect itself is highly variable, as the reviewer notes in the comment “…GluA2 increases are variable across cultures…” Presumably, if GluA2 increases are the mechanism of the mEPSC amplitude increases, then variable GluA2 increases should correlate with variable mEPSC amplitude increases, but that is not what we observed. We are left with the explanation that the immunofluorescence method itself is very variable. We have added the point to the Discussion, which reads, “the variability could arise from technical factors associated with the immunofluorescence method, and the mechanism of Rab3A-dependent homeostatic plasticity could be presynaptic for the NMJ and postsynaptic for cortical neurons.”

Finally, the implication of “Shouldn’t this be an indicator that something unusual has happened in this culture?” if it is not due to culture to culture variability in the homeostatic response itself, is that there was a technical problem with accurately measuring receptor levels. We have no reason to suspect anything was amiss in this set of coverslips (the values for controls and for TTX-treated were not outside the range of values in other experiments). In any of the coverslips, there may be variability in the amount of primary anti-GluA2 antibody, as this was added directly to the culture rather than prepared as a diluted solution and added to all the coverslips. But to remove this one experiment because it did not give the expected result is to allow bias to direct our data selection.

The authors further cite a study with comparable sample sizes, which shows a similar mismatch based on p values (Xu and Pozzo-Miller 2007), yet the effect sizes in this study actually match quite well (both ~160%). P values cannot be used to show whether two effects match, but effect sizes can. Therefore, the statement in lines 411-413 "... consistently leads to an increase in mEPSC amplitudes, and sometimes leads to an increase in synaptic GluA2 receptor cluster size" is not very convincing, and can hardly be used to support "the idea that there are additional sources contributing to the homeostatic increase in quantal size.”

We have the same situation; our effect sizes match (19.7% increase for mEPSC amplitude; 18.1% increase for GluA2 receptor cluster size, see Table 1), but in our case, the p value for receptors does not reach statistical significance. Our point here is that there is published evidence that the variability in receptor measurements is greater than the variability in electrophysiological measurements. But we have softened this point, removing the sentences containing “…consistently leads and sometimes...” and “……additional sources contributing…”.

I would suggest simply showing mEPSC and immunostaining data from all cultures in this experiment as additional evidence for homeostatic synaptic plasticity in WT cultures, and leave out the argument for "mismatch". The presynaptic location of Rab3A is sufficient to speculate a presynaptic regulation of this form of homeostatic compensation.

We have removed all uses of the word “mismatch,” but feel the presentation of the 3 matched experiments, 23-24 cells (Figure 5A, D), and the additional 2 experiments for a total of 5 cultures, 48-49 cells (Figure 5C, F), is important in order to demonstrate that the lack of statistically significant receptor response is due neither to a variable homeostatic response in the mEPSC amplitudes, nor to a small number of cultures.

Minor concerns:(1) Line 214, I see the authors cite literature to argue that GluA2 can form homomers and can conduct currents. While GluA2 subunits edited at the Q/R site (they are in nature) can form homomers with very low efficiency in exogenous systems such as HEK293 cells (as done in the cited studies), it's unlikely for this to happen in neurons (they can hardly traffic to synapses if possible at all).

We were unable to identify a key reference that characterized GluA2 homomers vs. heteromers in native cortical neurons, but we have rewritten the section in the manuscript to acknowledge the low conductance of homomers:

“…to assess whether GluA2 receptor expression, which will identify GluA2 homomers and GluA2 heteromers the former unlikely to contribute to mEPSCs given their low conductance relative to heteromers (Swanson et al., 1997; Mansour et al., 2001)…”

(2) Lines 221-222, the authors may have misinterpreted the results in Turrigiano 1998. This study does not show that the increase in receptors is most dramatic in the apical dendrite, in fact, this is the only region they have tested. The results in Figures 3b-c show that the effect size is independent of the distance from soma.

Figure 3 in Turrigiano et al., shows that the increase in glutamate responsiveness is higher at the cell body than along the primary dendrite. We have revised our description to indicate that an increase in responsiveness on the primary dendrite has been demonstrated in Turrigiano et al. 1998.

“We focused on the primary dendrite of pyramidal neurons as a way to reduce variability that might arise from being at widely ranging distances from the cell body, or, from inadvertently sampling dendritic regions arising from inhibitory neurons. In addition, it has been shown that there is a clear increase in response to glutamate in this region (Turrigiano et al., 1998).”

“…synaptic receptors on the primary dendrite, where a clear increase in sensitivity to exogenously applied glutamate was demonstrated (see Figure 3 in (Turrigiano et al., 1998)).

(3) Lines 309-310 (and other places mentioning TNFa), the addition of TNFa to this experiment seems out of place. The authors have not performed any experiment to validate the presence/absence of TNFa in their system (citing only 1 study from another lab is insufficient). Although it's convincing that glia Rab3A is not required for homeostatic plasticity here, the data does not suggest Rab3A's role (or the lack of) for TNFa in this process.

We have modified the paragraph in the Discussion that addresses the glial results, to describe more clearly the data that supported an astrocytic TNF-alpha mechanism: “TNF-alpha accumulates after activity blockade, and directly applied to neuronal cultures, can cause an increase in GluA1 receptors, providing a potential mechanism by which activity blockade leads to the homeostatic upregulation of postsynaptic receptors (Beattie et al., 2002; Stellwagen et al., 2005; Stellwagen and Malenka, 2006).”

We have also acknowledged that we cannot rule out TNF-alpha coming from neurons in the cortical cultures: “…suggesting the possibility that neuronal Rab3A can act via a non-TNF-alpha mechanism to contribute to homeostatic regulation of quantal amplitude, although we have not ruled out a neuronal Rab3A-mediated TNF-alpha pathway in cortical cultures.”

**Reviewer #3 (Public review):**
This manuscript presents a number of interesting findings that have the potential to increase our understanding of the mechanism underlying homeostatic synaptic plasticity (HSP). The data broadly support that Rab3A plays a role in HSP, although the site and mechanism of action remain uncertain.The authors clearly demonstrate that Rab3A plays a role in HSP at excitatory synapses, with substantially less plasticity occurring in the Rab3A KO neurons. There is also no apparent HSP in the Earlybird Rab3A mutation, although baseline synaptic strength is already elevated. In this context, it is unclear if the plasticity is absent, already induced by this mutation, or just occluded by a ceiling effect due to the synapses already being strengthened. Occlusion may also occur in the mixed cultures when Rab3A is missing from neurons but not astrocytes. The authors do appropriately discuss these options. The authors have solid data showing that Rab3A is unlikely to be active in astrocytes, Finally, they attempt to study the linkage between changes in synaptic strength and AMPA receptor trafficking during HSP, and conclude that trafficking may not be solely responsible for the changes in synaptic strength during HSP.Strengths:This work adds another player into the mechanisms underlying an important form of synaptic plasticity. The plasticity is likely only reduced, suggesting Rab3A is only partially required and perhaps multiple mechanisms contribute. The authors speculate about some possible novel mechanisms, including whether Rab3A is active pre-synaptically to regulate quantal amplitude.As Rab3A is primarily known as a pre-synaptic molecule, this possibility is intriguing. However, it is based on the partial dissociation of AMPAR trafficking and synaptic response and lacks strong support. On average, they saw a similar magnitude of change in mEPSC amplitude and GluA2 cluster area and integral, but the GluA2 data was not significant due to higher variability. It is difficult to determine if this is due to biology or methodology - the imaging method involves assessing puncta pairs (GluA2/VGlut1) clearly associated with a MAP2 labeled dendrite. This is a small subset of synapses, with usually less than 20 synapses per neuron analyzed, which would be expected to be more variable than mEPSC recordings averaged across several hundred events. However, when they reduce the mEPSC number of events to similar numbers as the imaging, the mESPC amplitudes are still less variable than the imaging data. The reason for this remains unclear. The pool of sampled synapses is still different between the methods and recent data has shown that synapses have variable responses during HSP. Further, there could be variability in the subunit composition of newly inserted AMPARs, and only assessing GluA2 could mask this (see below). It is intriguing that pre-synaptic changes might contribute to HSP, especially given the likely localization of Rab3A. But it remains difficult to distinguish if the apparent difference in imaging and electrophysiology is a methodological issue rather than a biological one. Stronger data, especially positive data on changes in release, will be necessary to conclude that pre-synaptic factors are required for HSP, beyond the established changes in post-synaptic receptor trafficking.

Regarding the concern that the lack of increase in receptors is due to a technical issue, please see General Response to Reviewers, above. We have also softened our conclusions throughout, acknowledging we cannot rule out a technical issue.

Other questions arise from the NASPM experiments, used to justify looking at GluA2 (and not GluA1) in the immunostaining. First, there is a strong frequency effect that is unclear in origin. One would expect NASPM to merely block some fraction of the post-synaptic current, and not affect pre-synaptic release or block whole synapses. But the change in frequency seems to argue (as the authors do) that some synapses only have CP-AMPARs, while the rest of the synapses have few or none. Another possibility is that there are pre-synaptic NASPM-sensitive receptors that influence release probability. Further, the amplitude data show a strong trend towards smaller amplitude following NASPM treatment (Fig 3B). The p value for both control and TTX neurons was 0.08 - it is very difficult to argue that there is no effect. The decrease on average is larger in the TTX neurons, and some cells show a strong effect. It is possible there is some heterogeneity between neurons on whether GluA1/A2 heteromers or GluA1 homomers are added during HSP. This would impact the conclusions about the GluA2 imaging as compared to the mEPSC amplitude data.

The key finding in Figure 3 is that NASPM did not eliminate the statistically significant increase in mEPSC amplitude after TTX treatment (Fig 3A). Whether or not NASPM sensitive receptors contribute to mESPC amplitude is a separate question (Fig 3B). We are open to the possibility that NASPM reduces mEPSC amplitude in both control and TTX treated cells (p = 0.08 for both), but that does not change our conclusion that NASPM has no effect on the TTX-induced increase in mEPSC amplitude. The mechanism underlying the decrease in mEPSC frequency following NASPM is interesting, but does not alter our conclusions regarding the role of Rab3A in homeostatic synaptic plasticity of mEPSC amplitude. In addition, the Reviewer does not acknowledge the Supplemental Figure #1, which shows a similar lack of correspondence between homeostatic increases in mEPSC amplitude and GluA1 receptors in two cultures where matched data were obtained. Therefore, we do not think our lack of a robust increase in receptors can be explained by our failing to look at the relevant receptor.

To understand the role of Rab3A in HSP will require addressing two main issues:(1) Is Rab3A acting pre-synaptically, post-synaptically or both? The authors provide good evidence that Rab3A is acting within neurons and not astrocytes. But where it is acting (pre or post) would aid substantially in understanding its role. The general view in the field has been that HSP is regulated post-synaptically via regulation of AMPAR trafficking, and considerable evidence supports this view. More concrete support for the authors' suggestion of a pre-synaptic site of control would be helpful.

We agree that definitive evidence for a presynaptic role of Rab3A in homeostatic plasticity of mEPSC amplitudes in mouse cortical cultures requires demonstrating that loss of Rab3A in postsynaptic neurons does not disrupt the plasticity, whereas loss in presynaptic neurons does. Without these data, we can only speculate that the Rab3A-dependence of homeostatic plasticity of quantal size in cortical neurons may be similar to that of the neuromuscular junction, where it cannot be receptors. We have added to the Discussion that the mechanism of Rab3A regulation of homeostatic plasticity of quantal amplitude could different between cortical neurons and the neuromuscular junction (lines 448-450 in markup,). Establishing a way to co-culture Rab3A-/- and Rab3A+/+ neurons in ratios that would allow us to record from a Rab3A-/- neuron that has mainly Rab3A+/+ inputs (or vice versa) is not impossible, but requires either transfection or transgenic expression with markers that identify the relevant genotype, and will be the subject of future experiments.

(2): Rab3A is also found at inhibitory synapses. It would be very informative to know if HSP at inhibitory synapses is similarly affected. This is particularly relevant as at inhibitory synapses, one expects a removal of GABARs or a decrease in GABA release (ie the opposite of whatever is happening at excitatory synapses). If both processes are regulated by Rab3A, this might suggest a role for this protein more upstream in the signaling; an effect only at excitatory synapses would argue for a more specific role just at those synapses.

We agree with the Reviewer, that it is important to determine the generality of Rab3A function in homeostatic plasticity. Establishing the homeostatic effect on mIPSCs and then examining them in Rab3A-/- cultures is a large undertaking and will be the subject of future experiments.

**Recommendations for the authors:**

**Reviewer #1 (Recommendations for the authors):**
Minor (remaining) points:(1) The figure referenced in the first response to the reviewers (Figure 5G) does not exist.

We meant Figure 5F, which has been corrected in the current response.

(2) I recommend showing the data without binning (despite some overlap).

The box plot in Origin will not allow not binning, but we can make the bin size so small that for all intents and purposes, there is close to 1 sample in each bin. When we do this, the majority of data are overlapped in a straight vertical line. Previously described concerns were regarding the gaps in the data, but it should be noted that these are cell means and we are not depicting the distributions of mEPSC amplitudes within a recording or across multiple recordings.

(3) Please auto-scale all axes from 0 (e.g., Fig 1E, F).

We have rescaled all mEPSC amplitude axes in box plots to go from 0 (Figures 1, 2 and 6).

(4) Typo in Figure legend 3: "NASPM (20 um)" => uM

Fixed.

**Reviewer #2 (Recommendations for the authors):**
(1) Line 140, frequencies are reported in Hz while other places are in sec-1, while these are essentially the same, they should be kept consistent in writing.

All mEPSC frequencies have been changed to sec^-1^, except we have left “Hz” for repetitive stimulation and filtering.

(2) Paragraph starting from line 163 (as well as other places where multiple groups are compared, such as the occlusion discussion), the authors assessed whether there was a change in baseline between WT and mutant group by doing pairwise tests, this is not the right test. A two-way ANOVA, or at least a multivariant test would be more appropriate.

We have performed a two-way ANOVA, with genotype as one factor, and treatment as the other factor. The p values in Figures 1 and 2 have been revised to reflect p values from the post-hoc Tukey test on the specific interactions (for each particular genotype, TTX vs CON effects). The difference in the two WT strains, untreated, was not significant in the Post-Hoc Tukey test, and we have revised the text. The difference between the untreated WT from the Rab3A+/Ebd colony and the untreated Rab3AEbd/Ebd mutant was still significant in the Post-Hoc Tukey test, and this has replaced the Kruskal-Wallis test. The two-way ANOVA was also applied to the neuron-glia experiments and p values in Figure 6 adjusted accordingly.

(3) Relevant to the second point under minor concerns, I suggest this sentence be removed, as reducing variability and avoiding inhibitory projects are reasons good enough to restrict the analysis to the apical dendrites.

We have revised the description of the Turrigiano et al., 1998 finding from their Figure 3 and feel it still strengthens the justification for choosing to analyze only synapses on the apical dendrite.

**Reviewer #3 (Recommendations for the authors):**
Minor points:The comments on lines 256-7 could seem misleading - the NASPM results wouldn't rule out contribution of those other subunits, only non-GluA2 containing combinations of those subunits. I would suggest revising this statement. Also, NASPM does likely have an effect, just not one that changes much with TTX treatment.

At new line 213 (markup) we have added the modifier “homomeric” to clarify our point that the lack of NASPM effect on the increase in mEPSC amplitude after TTX indicates that the increase is not due to more homomeric Ca^2+^-permeable receptors. We have always stated that NASPM reduces mEPSC amplitude, but it is in both control and treated cultures.

Strong conclusions based on a single culture (lines 314-5) seem unwarranted.

We have softened this statement with a “suggesting that” substituted for the previous “Therefore,” but stand by our point that the mEPSC amplitude data support a homeostatic effect of TTX in Culture #3, so the lack of increase in GluA2 cluster size needs an explanation other than variability in the homeostatic effect itself.

Saying (line 554) something is 'the only remaining possibility' also seems unwarranted.

We have softened this statement to read, “A remaining possibility…”.

Beattie EC, Stellwagen D, Morishita W, Bresnahan JC, Ha BK, Von Zastrow M, Beattie MS, Malenka RC (2002) Control of synaptic strength by glial TNFalpha. Science 295:2282-2285.

Mansour M, Nagarajan N, Nehring RB, Clements JD, Rosenmund C (2001) Heteromeric AMPA receptors assemble with a preferred subunit stoichiometry and spatial arrangement. Neuron 32:841-853. Stellwagen D, Malenka RC (2006) Synaptic scaling mediated by glial TNF-alpha. Nature 440:1054-1059.

Stellwagen D, Beattie EC, Seo JY, Malenka RC (2005) Differential regulation of AMPA receptor and GABA receptor trafficking by tumor necrosis factor-alpha. J Neurosci 25:3219-3228.

Swanson GT, Kamboj SK, Cull-Candy SG (1997) Single-channel properties of recombinant AMPA receptors depend on RNA editing, splice variation, and subunit composition. J Neurosci 17:5869.

Turrigiano GG, Leslie KR, Desai NS, Rutherford LC, Nelson SB (1998) Activity-dependent scaling of quantal amplitude in neocortical neurons. Nature 391:892-896.

Wang X, Wang Q, Yang S, Bucan M, Rich MM, Engisch KL (2011) Impaired activity-dependent plasticity of quantal amplitude at the neuromuscular junction of Rab3A deletion and Rab3A earlybird mutant mice. J Neurosci 31:3580-3588.

Wang X, Li Y, Engisch KL, Nakanishi ST, Dodson SE, Miller GW, Cope TC, Pinter MJ, Rich MM (2005) Activity-dependent presynaptic regulation of quantal size at the mammalian neuromuscular junction in vivo. J Neurosci 25:343-351.